# SynBench: Task-Agnostic Benchmarking of Pretrained Representations using Synthetic Data

## Abstract

Recent success in fine-tuning large models, that are pretrained on broad data at scale, on downstream tasks has led to a significant paradigm shift in deep learning, from task-centric model design to task-agnostic representation learning and task-specific fine-tuning. As the representations of pretrained models are used as a foundation for different downstream tasks, this paper proposes a new task-agnostic framework, *SynBench*, to measure the quality of pretrained representations using synthetic data. To address the challenge of task-agnostic data-free evaluation, we design synthetic binary classification proxy tasks with class conditional Gaussian mixtures to probe and compare model's robustness-accuracy performance on input synthetic data and their representations. Since the synthetic tasks spare access to real-life data, SynBench offers a holistic evaluation and informs the model designers of the intrinsic robustness level of the model given a user-specified threshold accuracy. Moreover, the use of class conditional Gaussian mixture allows us to derive a theoretically optimal robustness-accuracy tradeoff, which serves as a reference when evaluating the tradeoff on representations. By comparing the ratio of area-under-curve between the raw data and their representations, SynBench offers a quantifiable score for robustness-accuracy performance benchmarking. Our framework applies to a wide range of pretrained models taking continuous data inputs and is independent of the downstream tasks and datasets. Evaluated with several pretrained vision transformer models, the experimental results show that our SynBench score well matches the actual linear probing performance of the pre-trained model when fine-tuned on downstream tasks. Moreover, our framework can be used to inform the design of robust linear probing on pretrained representations to mitigate the robustness-accuracy tradeoff in downstream tasks.

## 1 Introduction

In recent years, the use of large pretrained neural networks for efficient fine-tuning on downstream tasks has prevailed in many application domains such as vision, language, and speech. Instead of designing task-dependent neural network architectures for different downstream tasks, the current methodology focuses on the principle of task-agnostic pretraining and task-specific finetuning, which uses a neural network pretrained on a large-scale dataset (often in a self-supervised or unsupervised manner) to extract generic representations of the input data, which we call *pretrained representations* for simplicity. The pretrained representations are then used as a foundation (Bommasani et al., 2021) to solve downstream tasks by training a linear head (i.e., linear probing) on the data representations with the labels provided by a downstream dataset, or by simply employing zero-shot inference. Moreover, to handle multi-modal data, one can use a similar neural network architecture (e.g., transformer) for multi-modal data representation learning and alignment. Successful examples following this new machine learning paradigm include the GPT-3 language model (Brown et al., 2020), the vision transformer (Arnab et al., 2021), and the CLIP image-text model (Radford et al., 2021), to name a few.

As large pretrained models are shown to achieve state-of-the-art performance on a variety of downstream tasks with minimal fine-tuning, there is an intensified demand for using pretrained representations from a large model for efficient finetuning. When gauging the usefulness of a pretrained model, it is a convention to compare the accuracy on selected real-life tasks. However, this ap-

proach has two possible drawbacks: (1) if the underlying pretrained model has hidden risks, such as lacking robustness to adversarial examples, the standard accuracy cannot inform the risk as it does not correlate well (even worse, sometimes has negative correlation) with adversarial robustness (Su et al., 2018). Therefore, the trending practice of pretraining and fine-tuning also signifies immediate damage to all downstream tasks. (2) the implications suggested by any "better" results on specific datasets are subjective to the datasets used for evaluation and could be inconclusive when the evaluation datasets change (e.g. ViT-L/16 is reportedly performing worse than ViT-B/16 on 4 out of 27 linear probing tasks according to Radford et al. (2021), and is incompetent to ViT-B/16 on finetuned medical tasks (Okolo et al., 2022; Tummala et al., 2022)). Consequently, an ideal pretrained model should entail both good accuracy and adversarial robustness, and the level of goodness can be measured in a task-agnostic manner. To address this emerging challenge, we propose a novel framework named *SynBench* to evaluate the quality of pretrained representations, in terms of quantifying the tradeoff between standard accuracy and adversarial robustness to input perturbations. Specifically, SynBench uses synthetic data generated from a conditional Gaussian distribution to establish a reference characterizing the robustness-accuracy tradeoff based on the Bayes optimal linear classifiers. Then, SynBench obtains the representations of the same synthetic data from the pretrained model and compares them to the reference for performance benchmarking. Finally, we define the ratio of area-under-curves in robustness-accuracy characterization as a quantifiable metric of the quality of pretrained representations. The entire procedure of SynBench is illustrated in Figure 1.

Our SynBench framework features the following key advantages.

1. *Soundness*: We formalize the fundamental tradeoff in robustness and accuracy of the considered conditional Gaussian model and use this characterization as a reference to benchmark the quality of pretrained representations.

2. *Task-independence*: Since the pretraining of large models is independent of the downstream datasets and tasks (e.g., through self-supervised or unsupervised training on broad data at scale), the use of synthetic data in SynBench provides a task-agnostic approach to evaluating pretrained representations without the knowledge of downstream tasks and datasets.

3. *Completeness and privacy*: The flexibility of generating synthetic data (e.g., by adopting a different data sampling procedure) offers a good proxy towards a more comprehensive evaluation of pretrained representations when fine-tuned on different downstream datasets, especially in the scenario when the available datasets are not representative of the entire downstream datasets. Moreover, the use of synthetic data enables full control and simulation over data size and distribution, protects data privacy, and can facilitate model auditing and governance.

We highlight our **main contributions** as follows.

• We propose SynBench, a novel task-agnostic framework that uses synthetic data to evaluate the quality of pretrained representations. The evaluation process of SynBench is independent of the downstream datasets and tasks and it applies to any model taking continuous data inputs.

• Evaluated with several pretrained vision transformers, our experimental results show that the metric provided by SynBench well matches the model performance in terms of adversarial robustness and standard accuracy when finetuned on several downstream datasets. For example, SynBench-Score suggests that the Imagenet21k pretrained network (*ViT-B/16-in21k*) improves with finetuning on Imagenet1k (*ViT-B/16*), echoing with the higher CIFAR10 and CIFAR10-c linear probing accuracy of *ViT-B/16*.

• We show that SynBench can be used to inform the design and selection of the hyperparameters in robust linear probing to mitigate the robustness-accuracy tradeoff when fine-tuned on downstream datasets. For example, conducting $\epsilon$-robust linear probing with $\epsilon$ selected by SynBench-Score gives *ViT-B/16* $0.6\%$ increase in CIFAR10 accuracy and $1.3\%$ increase in CIFAR10-c accuracy.

## 2 RELATED WORK

**Pretrained models in vision.** In the past few years, much focus in the machine learning community has been shift to train representation networks capable of extracting features for a variety of downstream tasks with minimal fine-tuning. Nowadays, many common vision tasks are achieved with the assistant of good backbones, e.g. classifications (Yu et al., 2022; Wortsman et al., 2022; Foret et al., 2020; Xie et al., 2020; Dosovitskiy et al., 2020; Chen et al., 2020a), object detection (Redmon

Figure 1: Overview of our SynBench framework. Step 1: generate class conditional Gaussian and form the inputs to the pretrained model; Step 2: gather rendered representations; Step 3: measure the expected robustness bound under a range of threshold accuracy for both input raw data and their representations according to equation 2 and obtain the expected bound-threshold accuracy plot; Step 4: calculate SynBench score by the relative area under curve of the representations (area B) to the input data (area A + area B) in the expected bound-threshold accuracy plot.

& Farhadi, 2017; Liu et al., 2016), segmentation (Chen et al., 2017; Xie et al., 2021), etc. Among the popular backbones, vision transformers (ViT) (Dosovitskiy et al., 2020) have attracted enormous interest. ViTs stem from Transformers (Vaswani et al., 2017) and split an image into patches, which are then treated as tokens as for the original Transformers. We will exemplify the use of SynBench using several pretrained ViTs.

**Benchmarking pretrained models.** Since pretrained models are used as a foundation for different downstream tasks, it is central to transfer learning (Neyshabur et al., 2020; Pruksachatkun et al., 2020), and also tightly related to model generalization (Qiao et al., 2020; Carlucci et al., 2019). To benchmark the performance of a pretrained model, it is a convention to apply the pretrained model for a number of popular tasks and conduct linear probing on the representations (Chen et al., 2020b; Dosovitskiy et al., 2020; Chen et al., 2020a; 2021). Besides linear probing, evaluation frameworks have been proposed based on mutual information (Bachman et al., 2019) and minimum description length (MDL) (Blier & Ollivier, 2018; Voita & Titov, 2020), which are reliant on the label information of the downstream tasks and are hence task-specific. Moreover, recent work (Whitney et al., 2020) also discussed the sensitivity of validation accuracy (nonlinear probes) and MDL to evaluation dataset size, and proposed a variant of MDL and a sample complexity based quantifier that depends on the data distribution.

It was not until recently that more fundamental questions are brought up related to the pretrained models (Bommasani et al., 2021; Tran et al., 2022; Zhang & Ré, 2022). Lately, Bommasani et al. (2021) raised practical concerns about the homogenization incentivized by the scale of the pretraining. Although the homogenization might help in achieving competitive performance for some downstream tasks, the defects are also inherited by all these downstreams. On that account, a more careful study of the fundamentals of pretrained models is of paramount importance. Tran et al. (2022) was dedicated to explore the reliability of pretrained models by devising 10 types of tasks on 40 datasets in evaluating the desired aspect of reliability. Furthermore, it is pointed out by Zhang & Ré (2022) that pretrained models may not be robust to subpopulation or group shift as shown in 9 benchmarks. The adversarial robustness is benchmarked by authors of (Shao et al., 2021; Paul & Chen, 2022), where Paul & Chen (2022) demonstrated the superior robustness of ViTs through Imagenet and Shao et al. (2021) conducted white-box and transfer attacks on Imagenet and CIFAR10.

**Optimal representations.** In the seminal work of deep representation theory, Achille & Soatto (2018) depicted the desired optimal representations in supervised learning to be sufficient for downstream task, invariant to the effect of nuisances, maximally disentangled, and has minimal mutual information between representations and inputs. Focusing more on generalization than compression, Dubois et al. (2020) gave the optimal representation based on $\mathcal{V}$-information (Xu et al., 2019) and probed generalization in deep learning. More recently, Ruan et al. (2021) defined the optimal representations for domain generalization. In (Dubois et al., 2022), authors characterize the idealized representation properties for invariant self-supervised representation learning. Specifically, idealized representation should be well-distinguished by the desired family of probes for potential invariant tasks, have sufficiently large dimension, and be invariant to input augmentations.

SynBench differs from the above quantifiers as it does not need knowledge of any downstream data and has controls over the evaluation set size since we could draw arbitrary number of synthetic data.

With the assumed synthetic data distribution, we could theoretically characterize the robustness-accuracy tradeoff that is independent to the downstream tasks. Therefore, SynBench provides a predefined standard of the tradeoff, which serves as the reference for representations induced by pretrained models. It should be also mentioned that, recently *sim-to-real* transfer paradigm has been leveraged to test the quality of real data, by projecting those onto the space of a model trained on large-scale synthetic data generated from a set of pre-defined grammar rules (Marzoev et al., 2020). SynBench, though conceptually similar at a very high level, is different from that line of work – as the focus of this work is to quantify the accuracy-robustness tradeoff of pretrained representations using synthetic data from conditional distributions.

## 3   SYNBENCH: METHODOLOGY AND EVALUATION

Without the knowledge of the downstream tasks and data, we aim to develop a task-agnostic framework to evaluate some fundamental behaviors of the representation network. As robustness is a key desired property, we probe the network to check how representation networks are preserving robustness in the original data. It is crucial to note that the probing method developed herein specifies the robustness-accuracy tradeoff in the pretrained representations, can be used for understanding (and possible ranking) different pretrained networks.

On the whole, we want to measure the idealized robustness-accuracy tradeoff using synthetic data. By propagating the Gaussian realizations through different representation networks, we can also measure the robustness-accuracy tradeoff for representations. We start this section by giving the synthetic data and the corresponding optimal linear classifier of interest.

### 3.1   SYNTHETIC DATA AND OPTIMAL LINEAR CLASSIFIER

We consider imbalanced) binary classification problems with data pair $(x, y)$ generated from the mixture of two Gaussian distributions $P_{\mu_1, \mu_2, \Sigma}$, such that

$$x|y = 1 \sim \mathcal{N}(\mu_1, \Sigma), \ x|y = -1 \sim \mathcal{N}(\mu_2, \Sigma),$$

or equivalently, $\quad x - \dfrac{\mu_1 + \mu_2}{2}|y = 1 \sim \mathcal{N}(\tilde{\mu}, \Sigma), \ x - \dfrac{\mu_1 + \mu_2}{2}|y = -1 \sim \mathcal{N}(-\tilde{\mu}, \Sigma), \quad$ (1)

where $y \in \mathcal{C} = \{+1, -1\}$, $P(y = +1) = \tau$, $P(y = -1) = 1 - \tau$, and $\tilde{\mu} = \frac{\mu_1 - \mu_2}{2}$. We focus on the class-balanced case ($\tau = \frac{1}{2}$) and defer the imbalanced case to the Appendix D. When sampling from this idealized distribution, we eliminate the factor of data bias and can benchmark the robustness degradation in an ideal setting.

Let $\| \cdot \|_p$ denote the $\ell_p$ norm of a vector for any $p \geq 1$. For a given classifier $f$ and input $x$ with $f(x) = y$, where $y$ is the predicted label, it is not rational for the classifier to respond differently to $x + \delta$ than to $x$ for a small perturbation level measured by $\|\delta\|_p$, i.e. inconsistent top-1 prediction (Szegedy et al., 2013; Goodfellow et al., 2014). Therefore, the level of (adversarial) robustness for a classifier can be measured by the minimum magnitude of perturbation that causes misclassification, i.e. $\|\Delta\|_p := \min_{\delta: f(x+\delta) \neq f(x)} \|\delta\|_p$. For a generic function $f$, solving the optimization problem exactly is hard (Katz et al., 2017; Sinha et al., 2018). Luckily, one can readily solve for the optimization if $f$ is affine (Moosavi-Dezfooli et al., 2016).

In the following, we will exploit this point and consider the linear classifier that minimizes the robust classification error. An ideal candidate classifier for the class conditional Gaussian (equation 1) is specified by the robust Bayes optimal classifier (Bhagoji et al., 2019; Dobriban et al., 2020). Specifically, it is stated that the optimal robust classifier (with a robust margin $\epsilon$) for data generated from equation 1 is a linear classifier. We derive the following result as a direct application of the fact. To simplify the exposition, we focus on the $\ell_2$ norm in the remainder of this paper. We refer the readers to Appendix C for general $\ell_p$-norm results. We use "bound" to denote the minimal perturbation of a sample.

**Theorem 1.** *For any sample $x$, the optimal robust classifier $f_\epsilon$ for $P_{\mu_1, \mu_2, \Sigma}$ gives*

*(i) the bound (decision margin)* $\|\Delta\|_2 = \dfrac{|(x - \frac{\mu_1 + \mu_2}{2})^T \Sigma^{-1}(\tilde{\mu} - z_\Sigma(\tilde{\mu}))|}{\|\Sigma^{-1}(\tilde{\mu} - z_\Sigma(\tilde{\mu}))\|_2}$,

*(ii) the scaled bound* $\|\bar{\Delta}\|_2 = \dfrac{|(x - \frac{\mu_1 + \mu_2}{2})^T \Sigma^{-1}(\tilde{\mu} - z_\Sigma(\tilde{\mu}))|}{|\tilde{\mu}^T \Sigma^{-1}(\tilde{\mu} - z_\Sigma(\tilde{\mu}))|}$.

*For a sample $x \sim P_{\mu_1, \mu_2, \Sigma}$, it further gives*

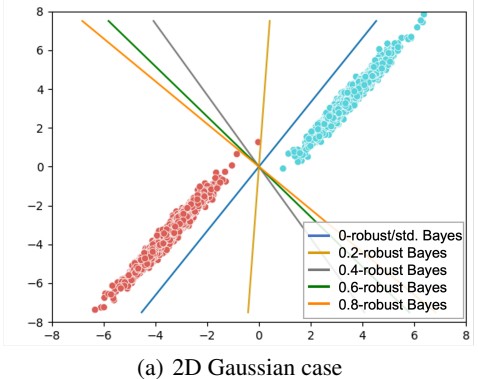
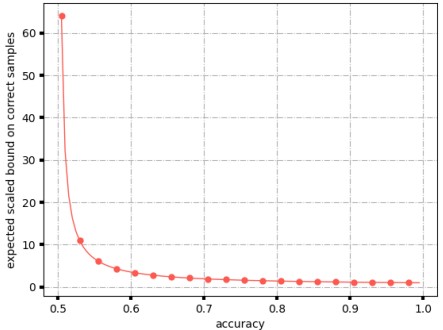

(a) 2D Gaussian case

(b) Theoretical robustness-accuracy tradeoff

Figure 2: Illustration of robustness-accuracy tradeoff suggested by $\epsilon$-robust Bayes optimal classifiers. Figure (a) depicts a class conditional 2D Gaussian case with decision boundaries drawn by $\epsilon$-robust Bayes optimal classifiers of varying $\epsilon$ values. Figure (b) draws the theoretically characterized robustness-accuracy tradeoff given in Theorem 1(iv).

*(iii) the standard accuracy $a = \Phi\left(\frac{\tilde{\mu}^T \Sigma^{-1}(\tilde{\mu} - z_\Sigma(\tilde{\mu}))}{\|\Sigma^{-1}(\tilde{\mu} - z_\Sigma(\tilde{\mu}))\|_\Sigma}\right)$,*

*(iv) the expected scaled bound $\mathbb{E}\left[\|\bar{\Delta}\|_2 \mid f_\epsilon(x) = y\right] = \frac{1}{\sqrt{2\pi}} \frac{1}{a \Phi^{-1}(a)} e^{-\frac{1}{2}\left(\Phi^{-1}(a)\right)^2} + 1$,*

*where $z_\Sigma$ is the solution of the convex problem $\arg\min_{\|z\|_2 \leq \epsilon}(\tilde{\mu} - z)^T \Sigma^{-1}(\tilde{\mu} - z)$ and $\Phi$ denotes the CDF of the standard normal distribution.*

We note that for samples drawn from $P_{\mu_1, \mu_2, \Sigma}$ and $\Sigma = \sigma^2 I_d$, all $\epsilon$-robust Bayes optimal classifier overlap with each other. For a general covariance matrix $\Sigma$, the $\epsilon$ of a $\epsilon$-robust Bayes classifier specifies the desired size of margin and demonstrates the robustness accuracy tradeoff. We give an illustrative 2D class conditional Gaussian example in Figure 2(a), where different $\epsilon$-robust Bayes classifiers give different overall margins at the cost of accuracy. Concretely, as $\epsilon$ increases, the robust Bayes optimal classifier rotates counterclockwise, leading to increased misclassifications, but also overall enlarged margins.

## 3.2 OBJECTIVE

For a given representation network parametrized by $\theta$, we are interested in evaluating the expected bounds on synthetic data and their representations, under a thresholding accuracy $a_t$, i.e. $\mathbb{E}_{\mu \sim \mathbb{P}_\mu, \Sigma \sim \mathbb{P}_\Sigma, x - \bar{\mu} \mid y \sim \mathcal{N}(y\mu, \Sigma)}\left[\|\bar{\Delta}\|_2 \mid f_\epsilon(x) = y, a > a_t\right]$ for $\bar{\Delta} = \bar{\Delta}_x$ and $\bar{\Delta}_z$ respectively, where $\mathbb{P}_\mu$ and $\mathbb{P}_\Sigma$ characterize the probability density function of the synthetic data manifold of interest, and $\bar{\mu}$ is a translation vector allowing non-symmetric class conditional Gaussian. Here, without the prior of applications, we assume $\mu = s \cdot 1_d/\sqrt{d}$, where $s$ denotes a random variable that follows uniform distribution and $1_d/\sqrt{d}$ is the normalized all-ones vector. For simplicity, we let $\Sigma = I_d$. Formally, we define $E_{\theta, \epsilon}(a_t)$ as

$$E_{\theta, \epsilon}(a_t) = \mathbb{E}_{s,x}\left[\|\bar{\Delta}\|_2 \mid f_\epsilon(x) = y, a(s, \epsilon) > a_t\right] = \frac{1}{n}\sum_i \mathbb{E}_x\left[\|\bar{\Delta}\|_2 \mid f_\epsilon(x) = y\right] \mathbb{1}_{a(s_i, \epsilon) > a_t}, \quad (2)$$

where $\mathbb{1}_{a(s_i, \epsilon) > a_t}$ is the indicator function specifying the $s_i, \epsilon$-dependent $a$ that surpasses the threshold accuracy $a_t$. We put the detailed derivation in Appendix A. In the following sections, we will illustrate how to calculate the inner expectation term $\mathbb{E}_x\left[\|\bar{\Delta}\|_2 \mid f_\epsilon(x) = y\right]$ for both the raw data and representations.

### 3.2.1 RAW DATA

For raw data synthesized from $P_{\mu_1, \mu_2, \Sigma}$ according to equation 1, the inner expectation term is given by Theorem 1(iv) $\mathbb{E}\left[\|\bar{\Delta}_x\|_2 \mid f_\epsilon(x) = y\right] = \frac{1}{\sqrt{2\pi}} \frac{1}{a \Phi^{-1}(a)} e^{-\frac{1}{2}\left(\Phi^{-1}(a)\right)^2} + 1$, where $a$ denotes the standard accuracy.

The subscript $x$ in the expected scaled bound $\mathbb{E}\left[\|\bar{\Delta}_x\|_2 \mid f_\epsilon(x) = y\right]$ indicates the raw data space, to distinguish from the scaled bound to be derived for representations. We highlight that Theorem 1(iv) directly gives a robustness-accuracy tradeoff. We plot the expected scaled bound as a function of accuracy in Figure 2(b). This tradeoff holds true when the data follow the conditional Gaussian exactly. In the proposed SynBench framework, we treat this theoretically-derived robustness-accuracy tradeoff as the reference, enabling a fair comparison among representations induced by different pretrained models.

### 3.2.2 REPRESENTATIONS

Given a pretrained network , we gather the representations of the Gaussian realizations and quantify the desired bound induced by robust Bayes optimal classifier in the representation space. When deriving the robust Bayes optimal classifier, we model the representations by a general conditional Gaussian $z|y = 1 \sim \mathcal{N}(\mu_1, \Sigma), z|y = -1 \sim \mathcal{N}(\mu_2, \Sigma)$. By Theorem 1(ii), we consider the optimal robust classifier for the modeled conditional Gaussian in the representation space to calculate the scaled bound $\|\bar{\Delta}_z\|_2 = \frac{|(z - \frac{\mu_1+\mu_2}{2})^T \Sigma^{-1}(\tilde{\mu} - z_\Sigma(\tilde{\mu}))|}{|\tilde{\mu}^T \Sigma^{-1}(\tilde{\mu} - z_\Sigma(\tilde{\mu}))|}$ for correctly-classified samples and the inner expectation is estimated empirically. It is worthwhile to note that now the Bayes optimal classifier does not necessarily coincide with robust Bayes optimal classifier even when we synthesized the dataset with an identity matrix covariance in the input space.

### 3.3 ROBUSTNESS-ACCURACY QUANTIFICATION OF REPRESENTATIONS

Recall that we aim to calculate

$$E_{\theta,\epsilon}(a_t) = \frac{1}{n}\sum_i \mathbb{E}_{x|y \sim \mathcal{N}(ys_i \cdot 1_d/\sqrt{d}, I_d)}\left[\|\bar{\Delta}\|_2 \mid f_\epsilon(x) = y\right]\mathbb{1}_{a(s_i,\epsilon) > a_t}$$

for both raw data and the representations (i.e. $\|\bar{\Delta}_x\|$ and $\|\bar{\Delta}_z\|$). We treat the expected bounds of the raw data under a threshold accuracy as the reference. Given a representation network, we compare the expected bounds of the representations rendered by representation networks with the reference.

We take $s \sim \mathcal{U}\{0.1, 5\}$ under the guidance of Theorem 1(iii). Specifically, as Theorem 1(iii) gives an analytical expected accuracy for class conditional Gaussian, we can obtain the desired range of $s$ by giving the accuracy. Now since we are interested in having the reference as a class conditional Gaussian that yields accuracy from 55% to almost 100%, we set the starting and ending $s$ by the fact that $\Phi(0.1) \sim 0.55$ and $\Phi(5) \sim 1.0$. We reiterate that with more accurate modelling of the data manifold of interest, SynBench can give more precise capture of the pretrained representation performance.

When the data is perfect Gaussian (e.g. input synthetic data), we calculate $E_{\theta,\epsilon}(a_t)$ with the help of Section 3.2.1. We note that $\bar{\Delta}_x$ is independent of pretrained network parameters $\theta$, and all the $\epsilon$-robust classifiers $f_\epsilon$ in the input space overlap with each other when $\Sigma = I_d$. We hereby denote the desired metric on the input synthetic data by $E(a_t)$, to distinguish from that on the representations $E_{\theta,\epsilon}(a_t)$. For representations, we calculate $E_{\theta,\epsilon}(a_t)$ with the help of Section 3.2.2 and the expectation is estimated empirically. We show an example of the probing results in Figure 3.

To integrate over all the desired threshold accuracy, we use the area under the curve (AUC) and give the ratio to the reference by

$$\text{SynBench-Score}(\theta, \epsilon, a_t) = \frac{\int_{a_t}^1 E_{\theta,\epsilon}(a)da}{\int_{a_t}^1 E(a)da}, \tag{3}$$

which correspond to $\frac{\text{area B}}{\text{area A + area B}}$ in Figure 3. Larger value of SynBench-Score implies better probing performance on pretrained representations.

## 4 EXPERIMENTAL RESULTS

In this experiment, we exemplify the use of SynBench given a pretrained representation network. In order to compare among network attributes, it is desirable to control the variates. In Table 1, we list severeal pretrained vision transformers (ViTs)[1](Dosovitskiy et al., 2020; Chen et al., 2021; Caron

---

[1]https://github.com/rwightman/pytorch-image-models

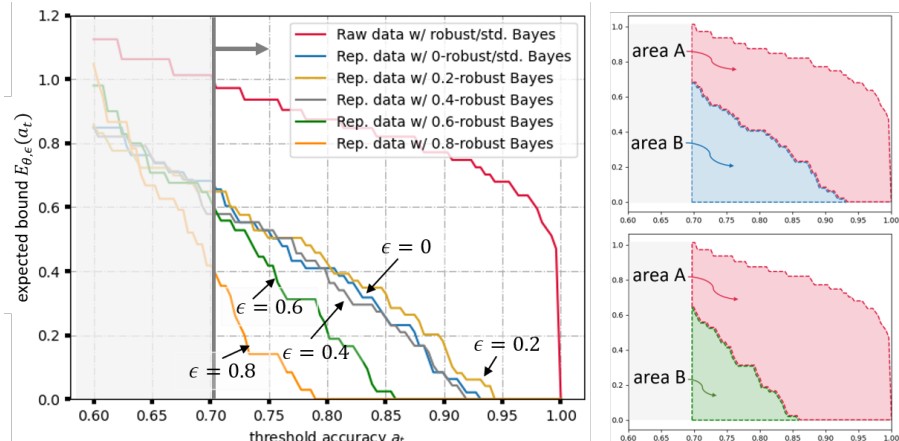

Figure 3: An example of the robustness-accuracy quantification of representations for ViT-B/16. (Left) The expected bound-threshold accuracy plot for the input raw data ($E(a_t)$) and representations ($E_{\theta,\epsilon}(a_t)$) with $\epsilon = 0 \sim 0.8$. (Right) The desired quantification SynBench-Score$(\theta, \epsilon, a_t) = \frac{\text{area B}}{\text{area A} + \text{area B}}$ (refer to equation 3) for $\epsilon = 0$ (top) and $\epsilon = 0.6$ (bottom).

| Model | Arch. | pretraining | fine-tuning | patch | # parameters (M) |
|---|---|---|---|---|---|
| ViT-Ti/16 | ViT-Tiny | Imgn21k | Imgn1k | 16 | 5.7 |
| ViT-B/16 | ViT-Base | Imgn21k | Imgn1k | 16 | 86.6 |
| ViT-B/16-in21k | ViT-Base | Imgn21k | No | 16 | 86.6 |
| ViT-B/32 | ViT-Base | Imgn21k | Imgn1k | 32 | 88.2 |
| ViT-L/16 | ViT-Large | Imgn21k | Imgn1k | 16 | 304.3 |
| ViT-S/16-DINO | ViT-Small | self-Imgn1k | No | 16 | 21.7 |
| ViT-S/8-DINO | ViT-Small | self-Imgn1k | No | 8 | 21.7 |
| ViT-B/16-DINO | ViT-Base | self-Imgn1k | No | 16 | 85.8 |
| ViT-B/8-DINO | ViT-Base | self-Imgn1k | No | 8 | 85.8 |
| Resnet50-SimCLRv2 | Resnet50 | self-Imgn1k | No | - | 144.4 |
| Resnet101-SimCLRv2 | Resnet101 | self-Imgn1k | No | - | 261.2 |
| Variation: | | | | | |
| Model size | ViT-{Ti,B,L}/16, ViT-{S,B}/16-DINO, ViT-{S,B}/8-DINO, Resnet{50,101}-SimCLRv2 | | | | |
| Finetuning | ViT-B/16{,-in21k} | | | | |
| ViT patch size | ViT-B/{16,32}, ViT-S/{16,32}-DINO, ViT-B/{16,32}-DINO | | | | |

Table 1: Model descriptions.

et al., 2021) and ResNets[2](Chen et al., 2020c), and make comparisons to our best knowledge. We note that the performance of these models might be nuanced by scheduler, curriculum, and training episodes, which are not captured in the above table. To provide a comprehensive evaluation, we give SynBench-Score$(\theta, \epsilon, a_t)$ with $a_t$ ranging from 0.7 to 0.9, and $\epsilon$ from 0 to 0.8. Due to space limit, some $a_t$ results are deferred to the appendix. The runtime of SynBench depends on the number of outcomes of the discrete uniform distribution $\mathcal{U}\{0.1, 5\}$. For one $s \sim \mathcal{U}\{0.1, 5\}$, it costs 59 seconds to generate 2048 Gaussian samples, 37 and 81 seconds to obtain the SynBench-Score for ViT-B/16 and ViT-L/16 on one GeForce RTX 2080 super. SynBench-Score offers a quantifiable score for robustness-accuracy performance benchmarking and is intrinsically a task-agnostic evaluation that characterizes general behaviors of the pretrained representations without the knowledge and use of any downstream data.

Apart from the task-agnostic metrics SynBench-Score developed in this paper, we also report linear probing accuracy on CIFAR10/ImageNet and CIFAR10-c/ImageNet-c (Hendrycks & Dietterich, 2019) to validate the standard and transfer accuracy (use the probing layer trained on CIFAR10/ImageNet to probe CIFAR10-c/ImageNet-c). We note that evaluating pretrained representations on real-life tasks is sensitive to the choice of tasks and the results may be inconclusive. For example, CIFAR10/ImageNet suggest that ViT-L performs better than ViT-B, wheras KITTI/SST (Geiger et al., 2012; Socher et al., 2013) (Radford et al., 2021, Table 10), FoodSeg103 (Wu et al., 2021, Table 8), X-ray images (Okolo et al., 2022, Table 4-8), magnetic resonance

---
[2]https://github.com/google-research/simclr

| $a_t = 0.7$ | $\epsilon = 0$ | $\epsilon = 0.2$ | $\epsilon = 0.4$ | $\epsilon = 0.6$ | $\epsilon = 0.8$ | CIFAR10 | CIFAR10-c | ImageNet | ImageNet-c |
|---|---|---|---|---|---|---|---|---|---|
| ViT-B/16 | 0.33 | 0.37 | 0.32 | 0.20 | 0.06 | 95.0 | 81.2 | 83.8 | 66.4 |
| ViT-B/16-in21k | 0.20 | 0.23 | 0.18 | 0.07 | 0.01 | 89.6 | 71.4 | 82.6 | 63.6 |

Table 2: Comparisons on the finetuning procedure in pretraining. The SynBench-Score of ViTs with or without finetuning pretrained representations, and the linear probing accuracy on CIFAR10/ImageNet and transfer accuracy on CIFAR10-c/ImageNet-c.

| $a_t$ | Model | $\epsilon = 0$ | $\epsilon = 0.2$ | $\epsilon = 0.4$ | $\epsilon = 0.6$ | $\epsilon = 0.8$ | CIFAR10 | CIFAR10-c | ImageNet | ImageNet-c |
|---|---|---|---|---|---|---|---|---|---|---|
| | ViT-Ti/16 | 0.01 | 0 | 0 | 0 | 0 | 81.9 | 59.1 | 74.8 | 43.3 |
| 0.7 | ViT-B/16 | 0.33 | 0.37 | 0.32 | 0.20 | 0.06 | 95.0 | 81.2 | 83.8 | 66.4 |
| | ViT-L/16 | 0.26 | 0.33 | 0.30 | 0.22 | 0.11 | 98.0 | 90.3 | 85.3 | 72.2 |
| | ViT-Ti/16 | 0 | 0 | 0 | 0 | 0 | 81.9 | 59.1 | 74.8 | 43.3 |
| 0.75 | ViT-B/16 | 0.26 | 0.30 | 0.25 | 0.11 | 0.01 | 95.0 | 81.2 | 83.8 | 66.4 |
| | ViT-L/16 | 0.19 | 0.27 | 0.24 | 0.16 | 0.04 | 98.0 | 90.3 | 85.3 | 72.2 |
| | ViT-Ti/16 | 0 | 0 | 0 | 0 | 0 | 81.9 | 59.1 | 74.8 | 43.3 |
| 0.8 | ViT-B/16 | 0.19 | 0.23 | 0.17 | 0.04 | 0 | 95.0 | 81.2 | 83.8 | 66.4 |
| | ViT-L/16 | 0.12 | 0.21 | 0.18 | 0.09 | 0 | 98.0 | 90.3 | 85.3 | 72.2 |
| | ViT-Ti/16 | 0 | 0 | 0 | 0 | 0 | 81.9 | 59.1 | 74.8 | 43.3 |
| 0.85 | ViT-B/16 | 0.10 | 0.15 | 0.09 | 0 | 0 | 95.0 | 81.2 | 83.8 | 66.4 |
| | ViT-L/16 | 0.05 | 0.13 | 0.10 | 0.03 | 0 | 98.0 | 90.3 | 85.3 | 72.2 |
| | ViT-Ti/16 | 0 | 0 | 0 | 0 | 0 | 81.9 | 59.1 | 74.8 | 43.3 |
| 0.9 | ViT-B/16 | 0.02 | 0.04 | 0.01 | 0 | 0 | 95.0 | 81.2 | 83.8 | 66.4 |
| | ViT-L/16 | 0 | 0.04 | 0.03 | 0 | 0 | 98.0 | 90.3 | 85.3 | 72.2 |

Table 3: Comparisons on the model sizes. The SynBench-Score of ViTs of different sizes, and the linear probing accuracy on CIFAR10/ImageNet and transfer accuracy on CIFAR10-c/ImageNet-c.

| $a_t = 0.7$ | $\epsilon = 0$ | $\epsilon = 0.2$ | $\epsilon = 0.4$ | $\epsilon = 0.6$ | $\epsilon = 0.8$ | CIFAR10 | CIFAR10-c | ImageNet | ImageNet-c |
|---|---|---|---|---|---|---|---|---|---|
| ViT-S/16-DINO | 0.48 | 0.47 | 0.42 | 0.32 | 0.17 | 95.3 | 75.9 | 75.3 | 47.5 |
| ViT-B/16-DINO | 0.55 | 0.58 | 0.53 | 0.46 | 0.35 | 96.5 | 78.9 | 76.4 | 52.1 |
| ViT-S/8-DINO | 0.40 | 0.42 | 0.39 | 0.34 | 0.26 | 96.2 | 78.0 | 79.0 | 53.9 |
| ViT-B/8-DINO | 0.50 | 0.56 | 0.50 | 0.40 | 0.30 | 97.0 | 80.6 | 79.5 | 53.7 |
| Res50-SimCLRv2 | 0.66 | 0.50 | 0.50 | 0.48 | 0.48 | 95.0 | 80.1 | 77.5 | 47.4 |
| Res101-SimCLRv2 | 0.60 | 0.64 | 0.55 | 0.51 | 0.48 | 95.6 | 80.9 | 78.7 | 50.1 |

Table 4: The SynBench-Score of self-supervised pretrained representations and the linear probing accuracy on CIFAR10/ImageNet and transfer accuracy on CIFAR10-c/ImageNet-c.

imaging (Tummala et al., 2022, Table 2-3) suggest the opposite. In contrast, because SynBench-Score is intrinsically a task-agnostic evaluation of the pretrained model, its result is independent of the choice of tasks.

**Fine-tuned pretraining representation.** When applying a pretrained representation network to the desired task, one can either only train a linear head on top of a fixed pretrained model, or perform fine-tuning of both the representation network and the linear head. Thus, in Table 2, we investigate how the fine-tuning process is affecting the representation networks. Specifically, both networks in Table 2 is pretrained on Imagenet 21k with supervision. After the pretraining, ViT-B/16 is further finetuned on Imagenet 1k. Interestingly, SynBench-Score shows that this finetuning is beneficial as improvements are witnessed across all $\epsilon$ with SynBench-Score, which well match the empirical observation give by CIFAR10 and CIFAR10-c and prior results (Kumar et al., 2021).

**Model size.** In Table 3, we compare ViTs of different sizes. Specifically, we perform SynBench on ViT-Ti, ViT-B, and ViT-L with patch size being 16. The model parameter $\theta$ is provided by the pretrained model. It is noticeable that ViT-B/16 is generally on par with ViT-L/16. When we set the threshold accuracy to be higher values, ViT-L/16 starts to give slightly better evaluations especially with larger $\epsilon$. One interesting observation is that for each model, SynBench-score is not necessarily monotonic in $\epsilon$, which indicates standard linear probing (i.e., $\epsilon = 0$) may not be the most effective way to probing pretrained representations in terms of robustness-accuracy performance, which is consistent with recent findings (Fan et al., 2021). See the "Robust linear probing" paragraph below for detailed analysis. We also observe that larger models exhibit better resilience (slower reduction in SynBench-score) as $\epsilon$ increases. On Self-supervised pretrained representations (Table 4), we observe that bigger models have higher SynBench-scores – ViT-B/16-DINO, ViT-B/16-DINO, and Res101-SimCLRv2 have bigger SynBench-scores compared to ViT-S/16-DINO, ViT-S/16-DINO, and Res50-SimCLRv2.

| $a_t = 0.6$ | $\epsilon = 0$ | $\epsilon = 0.2$ | $\epsilon = 0.4$ | $\epsilon = 0.6$ | $\epsilon = 0.8$ | CIFAR10 | CIFAR10-c | ImageNet | ImageNet-c |
|---|---|---|---|---|---|---|---|---|---|
| ViT-B/16 | 0.45 | 0.47 | 0.44 | 0.36 | 0.25 | 95.0 | 81.2 | 83.8 | 66.4 |
| ViT-B/32 | 0.02 | 0.03 | 0.03 | 0.01 | 0 | 92.2 | 76.6 | 80.5 | 61.4 |

Table 5: Comparisons on the ViT patch size. The SynBench-Score of ViTs of different patch sizes, and the linear probing accuracy on CIFAR10 and transfer accuracy on CIFAR10-c.

| | $\arg\max_\epsilon$ SynBench-Score | $\Delta$ robust linear probing mean (CIFAR10,CIFAR10-c) |
|---|---|---|
| ViT-Ti/16 | 0.1 | 70.5**-0.7** |
| ViT-B/16 | 0.2 | 88.1**+1.0** |
| ViT-L/16 | 0.2 | 94.2**+0.4** |
| ViT-B/16-in21k | 0.2 | 80.5**+0.4** |
| ViT-B/32 | 0.2 | 84.4**+0.8** |

Table 6: CIFAR10 and CIFAR10-c accuracy changes using $\epsilon$-robust linear probing with $\epsilon = \arg\max_\epsilon$ SynBench-Score.

**ViT patch size.** We also compare vision transformer patch sizes in Table 5. Specifically, we give ViT-B with patch size being 16 and 32, individually. SynBench-Scores show an consistent trend as the model performance on CIFAR10 and CIFAR10-c. From Table 4, we see that the SynBench-score of ViT-S/16-DINO is on par with that of ViT-S/8-DINO, and ViT-B/16-DINO has higher SynBench-score than ViT-B/8-DINO. Although linear probing on CIFAR10 and ImageNet do not share the trend, bigger patch size model (ViT-S/16-DINO) does perform better than smaller ones (ViT-S/8-DINO) on the PASCAL VOC (Everingham et al., 2010) segmentation task (Caron et al., 2021, Figure 4 bottom table).

**Robust linear probing.** According to Table 3, $0.2$-robust Bayes classifiers consistently give better scores compared to $0$-robust (standard) Bayes classifiers with ViT-B/16 and ViT-L/16. This offers us a quick way of gauging the suitable downstream robust probing parameter for the given pretrained model. We stipulate that observing a $0.2$-robust Bayes classifier to yield better SynBench-Score than a $0$-robust Bayes classifier may suggest the pretrained network to produce representations that have better overall performance with linear classifiers trained by $0.2$-robust linear probing. We validate this by performing robust linear probing on representations rendered by ViTs for CIFAR10 classifications. Results are shown in Table 6. For a given pretrained model, let $f$ and $g$ be the pretrained network and linear probing layer, we solve the optimization problem $\min_g \max_{\|\delta\| \le \epsilon} L(g(f(x+\delta)), y)$ using the PyTorch library Torchattacks[3] and 10-step PGDL2 attacks (Madry et al., 2018) for adversarial training. From Table 6, we see that robust linear probing with $\epsilon = \arg\max_\epsilon$ SynBench-Score generally gives a decent robustness-accuracy tradeoff. For example, with robust linear probing, we obtain a $0.6\%$ and $1.3\%$ increase in CIFAR10 standard and CIFAR10-c transfer accuracy with ViT-B/16 (as in Table 9). A more complete table on $\epsilon$-robust linear probing results with different $\epsilon$ is given in the appendix.

## 5 DISCUSSION AND CONCLUSION

In this paper, we propose a new **task-agnostic** framework *SynBench* for benchmarking the robustness-accuracy performance of pretrained representations. SynBench is fundamentally task-independent and provides a quantifiable score that does not reply on any real-life data. SynBench exploits an idealized data distribution, class conditional Gaussian mixture, to establish a theoretically-derived robustness-accuracy tradeoff, which serves as the reference for pretrained representations. Finally, a quantifiable score *SynBench-Score* is provided that compares the ratio of area-under-curve between the reference and the pretrained representations. We validate the usefulness of SynBench on several pretrained vision transformers in giving insightful comparisons on different model attributes (e.g. model size, fine-tuned pretraining representations, ViT patch size, linear probing).

While we delved into the robustness-accuracy performance of pretrained representations of vision transformers, we envision the SynBench framework to be further extended to other trustworthiness dimensions such as privacy, fairness, etc. Moreover, as the popularization of pretrained representations in various domains (e.g. vision, language, speech), we foresee SynBench to be generalized to more domains, and shed light on task-agnostic benchmarking designs.

---

[3]https://github.com/Harry24k/adversarial-attacks-pytorch

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

# A   OBJECTIVE

$$\begin{aligned}
E_{\theta,\epsilon}(a_t) &= \mathbb{E}_{s\sim\mathcal{U},x-\bar{\mu}|y\sim\mathcal{N}(\mu,\Sigma)}\left[\|\bar{\Delta}\|_2 \mid f_\epsilon(x)=y, a>a_t, \mu=s\cdot 1_d/\sqrt{d}, \Sigma=I_d\right] \\
&= \mathbb{E}_{s,x}\left[\|\bar{\Delta}\|_2 \mid f_\epsilon(x)=y, a(s,\epsilon)>a_t\right] \\
&= \sum_i \mathbb{E}_x\left[\|\bar{\Delta}\|_2 \mid f_\epsilon(x)=y, a(s_i,\epsilon)>a_t\right]\mathbb{P}(s=s_i) \\
&= \frac{1}{n}\sum_i \mathbb{E}_x\left[\|\bar{\Delta}\|_2 \mid f_\epsilon(x)=y, a(s_i,\epsilon)>a_t\right] \\
&= \frac{1}{n}\sum_i \mathbb{E}_x\left[\|\bar{\Delta}\|_2 \mid f_\epsilon(x)=y\right]\mathbb{1}_{a(s_i,\epsilon)>a_t}.
\end{aligned}$$

# B   PROOFS

**Theorem 1.** *For any sample $x$, the optimal robust classifier $f_\epsilon$ for $P_{\mu_1,\mu_2,\Sigma}$ gives*

(i) *the bound (decision margin)* $\|\Delta\|_2 = \frac{|(x-\frac{\mu_1+\mu_2}{2})^T\Sigma^{-1}(\tilde{\mu}-z_\Sigma(\tilde{\mu}))|}{\|\Sigma^{-1}(\tilde{\mu}-z_\Sigma(\tilde{\mu}))\|_2}$,

(ii) *the scaled bound* $\|\bar{\Delta}\|_2 = \frac{|(x-\frac{\mu_1+\mu_2}{2})^T\Sigma^{-1}(\tilde{\mu}-z_\Sigma(\tilde{\mu}))|}{|\tilde{\mu}^T\Sigma^{-1}(\tilde{\mu}-z_\Sigma(\tilde{\mu}))|}$.

*For a sample $x \sim P_{\mu_1,\mu_2,\Sigma}$, it further gives*

(iii) *the standard accuracy* $a = \Phi(\frac{\tilde{\mu}^T\Sigma^{-1}(\tilde{\mu}-z_\Sigma(\tilde{\mu}))}{\|\Sigma^{-1}(\tilde{\mu}-z_\Sigma(\tilde{\mu}))\|_\Sigma})$,

(iv) *the expected scaled bound* $\mathbb{E}\left[\|\bar{\Delta}\|_2 \mid f_\epsilon(x)=y\right] = \frac{1}{\sqrt{2\pi}}\frac{1}{a\Phi^{-1}(a)}e^{-\frac{1}{2}\left(\Phi^{-1}(a)\right)^2}+1$,

*where $z_\Sigma$ is the solution of the convex problem $\arg\min_{\|z\|_2\leq\epsilon}(\tilde{\mu}-z)^T\Sigma^{-1}(\tilde{\mu}-z)$ and $\Phi$ denotes the CDF of the standard normal distribution.*

*Proof.* (i) Following Bhagoji et al. (2019); Dan et al. (2020), the Bayes optimal robust classifier for the general non-symmetric conditional Gaussians $P_{\mu_1,\mu_2,\Sigma}$ specified in equation 1 is

$$f_\epsilon(x) = sign\left\{\left(x-\frac{\mu_1+\mu_2}{2}\right)^T\Sigma^{-1}\left(\tilde{\mu}-z_\Sigma(\tilde{\mu})\right)\right\}, \tag{4}$$

where $sign(\cdot)$ is the typical sign function and $z_\Sigma$ is the solution of the convex problem $\arg\min_{\|z\|_2\leq\epsilon}(\tilde{\mu}-z)^T\Sigma^{-1}(\tilde{\mu}-z)$. The corresponding decision boundary is at $\left((x+\delta)-\frac{\mu_1+\mu_2}{2}\right)^T\Sigma^{-1}\left(\tilde{\mu}-z_\Sigma(\tilde{\mu})\right)=0$,

$$\implies \quad \Delta = \arg\min\|\delta\|_2 \quad \text{s.t.} \quad \delta^T\Sigma^{-1}\left(\tilde{\mu}-z_\Sigma(\tilde{\mu})\right)=-\left(x-\frac{\mu_1+\mu_2}{2}\right)^T\Sigma^{-1}\left(\tilde{\mu}-z_\Sigma(\tilde{\mu})\right)$$

$$\implies \quad \|\Delta\|_2 = \frac{|(x-\frac{\mu_1+\mu_2}{2})^T\Sigma^{-1}(\tilde{\mu}-z_\Sigma(\tilde{\mu}))|}{\|\Sigma^{-1}(\tilde{\mu}-z_\Sigma(\tilde{\mu}))\|_2}.$$

(ii) Since the bound $\|\Delta\|_2$ is subject to the positions of two Gaussians, we scale the bound by the distance from Gaussian centers to the classifier, $\frac{|\tilde{\mu}^T\Sigma^{-1}(\tilde{\mu}-z_\Sigma(\tilde{\mu}))|}{\|\Sigma^{-1}(\tilde{\mu}-z_\Sigma(\tilde{\mu}))\|_2}$ and obtain

$$\begin{aligned}
\|\bar{\Delta}\|_2 &= \frac{|(x-\frac{\mu_1+\mu_2}{2})^T\Sigma^{-1}(\tilde{\mu}-z_\Sigma(\tilde{\mu}))|}{\|\Sigma^{-1}(\tilde{\mu}-z_\Sigma(\tilde{\mu}))\|_2}\frac{\|\Sigma^{-1}(\tilde{\mu}-z_\Sigma(\tilde{\mu}))\|_2}{|\tilde{\mu}^T\Sigma^{-1}(\tilde{\mu}-z_\Sigma(\tilde{\mu}))|} \\
&= \frac{|(x-\frac{\mu_1+\mu_2}{2})^T\Sigma^{-1}(\tilde{\mu}-z_\Sigma(\tilde{\mu}))|}{|\tilde{\mu}^T\Sigma^{-1}(\tilde{\mu}-z_\Sigma(\tilde{\mu}))|}.
\end{aligned}$$

(iii) For sample $x \sim P_{\mu_1, \mu_2, \Sigma}$, consider the Bayes optimal robust classifier in equation 4, we can calculate the analytical standard accuracy by

$$
\begin{aligned}
& \mathbb{P}(y=1)\mathbb{P}\left[f_\epsilon(x)=1 \mid y=1\right] + \mathbb{P}(y=-1)\mathbb{P}\left[f_\epsilon(x)=-1 \mid y=-1\right] \\
=& \mathbb{P}\left[f_\epsilon(x)=1 \mid y=1\right] \\
=& \mathbb{P}\left[(x-\frac{\mu_1+\mu_2}{2})^T\Sigma^{-1}(\tilde{\mu}-z_\Sigma(\tilde{\mu})) > 0 \mid y=1\right] \\
=& \mathbb{P}\left[(\tilde{\mu}+w)^T\Sigma^{-1}(\tilde{\mu}-z_\Sigma(\tilde{\mu})) > 0\right], \quad w \sim \mathcal{N}(0,\Sigma) \\
=& \mathbb{P}\left[w^T\Sigma^{-1}(\tilde{\mu}-z_\Sigma(\tilde{\mu})) > -\tilde{\mu}^T\Sigma^{-1}(\tilde{\mu}-z_\Sigma(\tilde{\mu}))\right], \quad w \sim \mathcal{N}(0,\Sigma) \\
=& \mathbb{P}\left[\frac{w^T\Sigma^{-1}(\tilde{\mu}-z_\Sigma(\tilde{\mu}))}{\|\Sigma^{-1}(\tilde{\mu}-z_\Sigma(\tilde{\mu}))\|_\Sigma} > -\frac{\tilde{\mu}^T\Sigma^{-1}(\tilde{\mu}-z_\Sigma(\tilde{\mu}))}{\|\Sigma^{-1}(\tilde{\mu}-z_\Sigma(\tilde{\mu}))\|_\Sigma}\right], \quad \frac{w^T\Sigma^{-1}(\tilde{\mu}-z_\Sigma(\tilde{\mu}))}{\|\Sigma^{-1}(\tilde{\mu}-z_\Sigma(\tilde{\mu}))\|_\Sigma} \sim \mathcal{N}(0,1) \\
=& \Phi(\frac{\tilde{\mu}^T\Sigma^{-1}(\tilde{\mu}-z_\Sigma(\tilde{\mu}))}{\|\Sigma^{-1}(\tilde{\mu}-z_\Sigma(\tilde{\mu}))\|_\Sigma}).
\end{aligned}
$$

(iv) For sample $x \sim P_{\mu_1,\mu_2,\Sigma}$, let $a$ denote the accuracy, $t$ denote $x - \frac{\mu_1+\mu_2}{2}$, and $w$ denote $\Sigma^{-1}(\tilde{\mu}-z_\Sigma(\tilde{\mu}))$. From (iii), we have that the standard accuracy of conditional Gaussian samples with the Bayes optimal (robust) classifier is $\Phi(\frac{\tilde{\mu}^T w}{\|w\|_\Sigma})$, so $\frac{\tilde{\mu}^T w}{\|w\|_\Sigma} = \Phi^{-1}(a)$. Since for binary classification, we only care about accuracy from 0.5 to 1, so we should have $\tilde{\mu}^T w > 0$.

Now consider the classifier in equation 4 and the corresponding scaled bound from (ii),

$$
\|\bar{\Delta}\|_2 = \frac{|(x-\frac{\mu_1+\mu_2}{2})^T\Sigma^{-1}(\tilde{\mu}-z_\Sigma(\tilde{\mu}))|}{|\tilde{\mu}^T\Sigma^{-1}(\tilde{\mu}-z_\Sigma(\tilde{\mu}))|} = \frac{|t^T w|}{|\tilde{\mu}^T w|} = \frac{|t^T w|}{\tilde{\mu}^T w}.
$$

Since $t|y \sim \mathcal{N}(y\tilde{\mu}, \Sigma)$, we have $t^T w|y \sim \mathcal{N}(y\tilde{\mu}^T w, w^T\Sigma^T w)$. When we only want to get the expected scaled bound of the correctly-classified samples, we have that

$$
\begin{aligned}
\mathbb{E}\left[\|\bar{\Delta}\|_2 \mid f_\epsilon(x)=y\right] &= \frac{1}{\tilde{\mu}^T w}\mathbb{E}\left[|t^T w| \mid f_\epsilon(x)=y\right] \\
&= \frac{1}{2\tilde{\mu}^T w}\mathbb{E}\left[|t^T w| \mid f_\epsilon(x)=y=1\right] + \frac{1}{2\tilde{\mu}^T w}\mathbb{E}\left[|t^T w| \mid f_\epsilon(x)=y=-1\right] \\
&= \frac{1}{2\tilde{\mu}^T w}\mathbb{E}\left[t^T w \mid y=1, t^T w \geq 0\right] + \frac{1}{2\tilde{\mu}^T w}\mathbb{E}\left[-t^T w \mid y=-1, t^T w < 0\right].
\end{aligned}
$$

Recall that $t^T w|y \sim \mathcal{N}(y\tilde{\mu}^T w, w^T\Sigma^T w)$, then by the mean of truncated normal distribution, it is true that

$$
\begin{aligned}
\mathbb{E}\left[t^T w \mid y=1, t^T w \geq 0\right] &= \tilde{\mu}^T w + \sqrt{w^T\Sigma^T w}\frac{\phi(\frac{0-\tilde{\mu}^T w}{\sqrt{w^T\Sigma^T w}})}{1-\Phi(\frac{0-\tilde{\mu}^T w}{\sqrt{w^T\Sigma^T w}})} \\
&= \tilde{\mu}^T w + \sqrt{w^T\Sigma^T w}\frac{\phi(-\frac{\tilde{\mu}^T w}{\sqrt{w^T\Sigma^T w}})}{1-\Phi(-\frac{\tilde{\mu}^T w}{\sqrt{w^T\Sigma^T w}})} \\
&= \tilde{\mu}^T w + \sqrt{w^T\Sigma^T w}\frac{1}{\sqrt{2\pi}\Phi(\frac{\tilde{\mu}^T w}{\sqrt{w^T\Sigma^T w}})}e^{-\frac{1}{2}\left(\frac{\tilde{\mu}^T w}{\sqrt{w^T\Sigma^T w}}\right)^2} \\
\mathbb{E}\left[-t^T w \mid y=-1, t^T w < 0\right] &= -\mathbb{E}\left[t^T w \mid y=-1, t^T w < 0\right] \\
&= -\left(-\tilde{\mu}^T w - \sqrt{w^T\Sigma^T w}\frac{\phi(\frac{0+\tilde{\mu}^T w}{\sqrt{w^T\Sigma^T w}})}{\Phi(\frac{0+\tilde{\mu}^T w}{\sqrt{w^T\Sigma^T w}})}\right) \\
&= \tilde{\mu}^T w + \sqrt{w^T\Sigma^T w}\frac{1}{\sqrt{2\pi}\Phi(\frac{\tilde{\mu}^T w}{\sqrt{w^T\Sigma^T w}})}e^{-\frac{1}{2}\left(\frac{\tilde{\mu}^T w}{\sqrt{w^T\Sigma^T w}}\right)^2}.
\end{aligned}
$$

Therefore

$$\mathbb{E}\left[\|\bar{\Delta}\|_2 \mid f_\epsilon(x) = y\right] = \frac{1}{\tilde{\mu}^T w}\left(\tilde{\mu}^T w + \sqrt{w^T \Sigma^T w}\frac{1}{\sqrt{2\pi}\Phi(\frac{\tilde{\mu}^T w}{\sqrt{w^T \Sigma^T w}})}e^{-\frac{1}{2}\left(\frac{\tilde{\mu}^T w}{\sqrt{w^T \Sigma^T w}}\right)^2}\right)$$

$$= 1 + \frac{\sqrt{w^T \Sigma^T w}}{\tilde{\mu}^T w}\frac{1}{\sqrt{2\pi}\Phi(\frac{\tilde{\mu}^T w}{\sqrt{w^T \Sigma^T w}})}e^{-\frac{1}{2}\left(\frac{\tilde{\mu}^T w}{\sqrt{w^T \Sigma^T w}}\right)^2}.$$

By replacing $\frac{\tilde{\mu}^T w}{\sqrt{w^T \Sigma^T w}}$ by $\Phi^{-1}(a)$, we got

$$\mathbb{E}\left[\|\bar{\Delta}\|_2 \mid f_\epsilon(x) = y\right] = \frac{1}{\sqrt{2\pi}}\frac{1}{a\Phi^{-1}(a)}e^{-\frac{1}{2}\left(\Phi^{-1}(a)\right)^2} + 1.$$

$\square$

## C  GENERAL $\ell_p$ RESULTS

We note that our results in Appendix B can be straightforwardly generalized to $\ell_p$. Given an $\ell_p$ adversarial budget $\epsilon$:

**Theorem 2.** *For any sample $x$, the optimal robust classifier $f_\epsilon$ for $P_{\mu_1,\mu_2,\Sigma}$ gives*

(i) *the bound (decision margin) $\|\Delta\|_p = \frac{|(x - \frac{\mu_1 + \mu_2}{2})^T \Sigma^{-1}(\tilde{\mu} - z_\Sigma(\tilde{\mu}))|}{\|\Sigma^{-1}(\tilde{\mu} - z_\Sigma(\tilde{\mu}))\|_q}$,*

(ii) *the scaled bound $\|\bar{\Delta}\|_p = \frac{|(x - \frac{\mu_1 + \mu_2}{2})^T \Sigma^{-1}(\tilde{\mu} - z_\Sigma(\tilde{\mu}))|}{|\tilde{\mu}^T \Sigma^{-1}(\tilde{\mu} - z_\Sigma(\tilde{\mu}))|}$.*

*For sample $x \sim P_{\mu_1,\mu_2,\Sigma}$, it further gives*

(iii) *the standard accuracy $a = \Phi(\frac{\tilde{\mu}^T \Sigma^{-1}(\tilde{\mu} - z_\Sigma(\tilde{\mu}))}{\|\Sigma^{-1}(\tilde{\mu} - z_\Sigma(\tilde{\mu}))\|_\Sigma})$,*

(iv) *the expected scaled bound $\mathbb{E}\left[\|\bar{\Delta}\|_p \mid f_\epsilon(x) = y\right] = \frac{1}{\sqrt{2\pi}}\frac{1}{a\Phi^{-1}(a)}e^{-\frac{1}{2}\left(\Phi^{-1}(a)\right)^2} + 1$,*

*where $z_\Sigma$ is the solution of the convex problem $\arg\min_{\|z\|_p \leq \epsilon}(\tilde{\mu} - z)^T \Sigma^{-1}(\tilde{\mu} - z)$ and $\Phi$ denotes the CDF of the standard normal distribution.*

*Proof.* We follow the proof of Theorem 1 and consider the classifier in equation 4. By Hölder's inequality, we now have the corresponding lower bound and scaled lower bound as

$$\|\Delta\|_p = \frac{|(x - \frac{\mu_1 + \mu_2}{2})^T \Sigma^{-1}(\tilde{\mu} - z_\Sigma(\tilde{\mu}))|}{\|\Sigma^{-1}(\tilde{\mu} - z_\Sigma(\tilde{\mu}))\|_q}$$

$$\|\bar{\Delta}\|_p = \frac{|(x - \frac{\mu_1 + \mu_2}{2})^T \Sigma^{-1}(\tilde{\mu} - z_\Sigma(\tilde{\mu}))|}{\|\Sigma^{-1}(\tilde{\mu} - z_\Sigma(\tilde{\mu}))\|_q}\frac{\|\Sigma^{-1}(\tilde{\mu} - z_\Sigma(\tilde{\mu}))\|_q}{|\tilde{\mu}^T \Sigma^{-1}(\tilde{\mu} - z_\Sigma(\tilde{\mu}))|}$$

$$= \frac{|(x - \frac{\mu_1 + \mu_2}{2})^T \Sigma^{-1}(\tilde{\mu} - z_\Sigma(\tilde{\mu}))|}{|\tilde{\mu}^T \Sigma^{-1}(\tilde{\mu} - z_\Sigma(\tilde{\mu}))|},$$

where $\frac{1}{p} + \frac{1}{q} = 1$. The remainder of the proof will then follows as in Theorem 1. $\square$

**Remark.** In general, in the case that $\Sigma$ is singular, we can apply the economy-size (thin) decomposition with nonzero eigenvalues $\Sigma = F\Lambda F^T$. Then, with a general non-symmetric conditional Gaussians

$$x|y = 1 \sim \mathcal{N}(\mu_1, \Sigma), \ x|y = -1 \sim \mathcal{N}(\mu_2, \Sigma),$$

we apply proper translation to symmetric conditional Gaussians

$$F^T x|y = 1 \sim \mathcal{N}(F^T \mu_1, \Sigma), \ F^T x|y = -1 \sim \mathcal{N}(F^T \mu_2, \Sigma),$$

$$F^T x - F^T \frac{\mu_1 + \mu_2}{2}|y = 1 \sim \mathcal{N}(\tilde{\mu}, \Sigma), \ F^T x - F^T \frac{\mu_1 + \mu_2}{2}|y = -1 \sim \mathcal{N}(-\tilde{\mu}, \Sigma),$$

where $\tilde{\mu} = F^T \frac{\mu_1 - \mu_2}{2}$.

# D  CLASS IMBALANCE

Given an $\ell_2$ adversarial budget $\epsilon \leq \|\mu\|_2$, consider the conditional Gaussian in equation 1 with $\Sigma = I_d$ ($d$ by $d$ identity matrix) and general class prior $\tau$, then the following theorem holds.

**Theorem 3.** *For any sample $x$, the optimal robust classifier $f_\epsilon$ for $P_{\mu_1,\mu_2,I_d}$ gives*

(i) *the bound (decision margin)* $\|\Delta\|_2 = \frac{|(x-\frac{\mu_1+\mu_2}{2})^T\tilde{\mu}(1-\epsilon/\|\tilde{\mu}\|_2)-q/2|}{\|\tilde{\mu}(1-\epsilon/\|\tilde{\mu}\|_2)\|_2}$,

(ii) *the scaled bound* $\|\bar{\Delta}\|_2 = \frac{|(x-\frac{\mu_1+\mu_2}{2})^T\tilde{\mu}(1-\epsilon/\|\tilde{\mu}\|_2)-q/2|}{|\tilde{\mu}^T\tilde{\mu}(1-\epsilon/\|\tilde{\mu}\|_2)-q/2|}$.

*For a sample $x \sim P_{\mu_1,\mu_2,I_d}$, it further gives*

(iii) *the standard accuracy* $a = \tau\Phi(\frac{\tilde{\mu}^T w-q/2}{\|w\|_2}) + (1-\tau)\Phi(\frac{\tilde{\mu}^T w+q/2}{\|w\|_2})$,

(iv) *the expected scaled bound* $\mathbb{E}\left[\|\bar{\Delta}\|_2 \mid f_\epsilon(x) = y\right] =$

$$\frac{\tau}{\tilde{\mu}^T w - q/2}\left(\tilde{\mu}^T w - q/2 + \|w\|_2\frac{\phi(\frac{-\tilde{\mu}^T w+q/2}{\|w\|_2})}{\Phi(\frac{\tilde{\mu}^T w-q/2}{\|w\|_2})}\right) + \frac{1-\tau}{\tilde{\mu}^T w - q/2}\left(\tilde{\mu}^T w + q/2 + \|w\|_2\frac{\phi(\frac{\tilde{\mu}^T w+q/2}{\|w\|_2})}{\Phi(\frac{\tilde{\mu}^T w+q/2}{\|w\|_2})}\right).$$

*where $q = ln\{(1-\tau)/\tau\}$, $w = \tilde{\mu}(1-\epsilon/\|\tilde{\mu}\|_2)$, $\phi$ and $\Phi$ denotes the PDF and CDF of the standard normal distribution.*

*Proof.* (i) Consider the Bayes optimal $\ell_2$ $\epsilon$-robust classifier (Dobriban et al., 2020, Theorem 4.1)

$$f_\epsilon(x) = sign\left\{\left(x - \frac{\mu_1 + \mu_2}{2}\right)^T\tilde{\mu}(1 - \epsilon/\|\tilde{\mu}\|_2) - q/2\right\}, \tag{5}$$

where $q = ln\{(1 - \tau)/\tau\}$. For any $x$,

$$\|\Delta\|_2 = \frac{|(x - \frac{\mu_1+\mu_2}{2})^T\tilde{\mu}(1 - \epsilon/\|\tilde{\mu}\|_2) - q/2|}{\|\tilde{\mu}(1 - \epsilon/\|\tilde{\mu}\|_2)\|_2}.$$

(ii) Since the bound $\|\Delta\|_2$ is subject to the positions of two Gaussians, we scale the bound by the distance from Gaussian centers to the classifier, $\frac{|\tilde{\mu}^T\tilde{\mu}(1-\epsilon/\|\tilde{\mu}\|_2)-q/2|}{\|\tilde{\mu}(1-\epsilon/\|\tilde{\mu}\|_2)\|_2}$ and obtain

$$\|\bar{\Delta}\|_2 = \frac{|(x - \frac{\mu_1+\mu_2}{2})^T\tilde{\mu}(1 - \epsilon/\|\tilde{\mu}\|_2) - q/2|}{\|\tilde{\mu}(1 - \epsilon/\|\tilde{\mu}\|_2)\|_2}\frac{\|\tilde{\mu}(1 - \epsilon/\|\tilde{\mu}\|_2)\|_2}{|\tilde{\mu}^T\tilde{\mu}(1 - \epsilon/\|\tilde{\mu}\|_2) - q/2|}$$

$$= \frac{|(x - \frac{\mu_1+\mu_2}{2})^T\tilde{\mu}(1 - \epsilon/\|\tilde{\mu}\|_2) - q/2|}{|\tilde{\mu}^T\tilde{\mu}(1 - \epsilon/\|\tilde{\mu}\|_2) - q/2|}.$$

(iii) For sample $x \sim P_{\mu_1, \mu_2, I_d}$, consider the Bayes optimal robust classifier in equation 4, we can calculate the analytical standard accuracy by

$$\begin{aligned}
&\mathbb{P}(y=1)\mathbb{P}\left[f_\epsilon(x) = 1 \mid y = 1\right] + \mathbb{P}(y=-1)\mathbb{P}\left[f_\epsilon(x) = -1 \mid y = -1\right] \\
=&\tau\mathbb{P}\left[f_\epsilon(x) = 1 \mid y = 1\right] + (1-\tau)\left[f_\epsilon(x) = -1 \mid y = -1\right] \\
=&\tau\mathbb{P}\left[(x - \frac{\mu_1 + \mu_2}{2})^T\tilde{\mu}(1 - \epsilon/\|\tilde{\mu}\|_2) - q/2 > 0 \mid y = 1\right] \\
&+ (1-\tau)\mathbb{P}\left[(x - \frac{\mu_1 + \mu_2}{2})^T\tilde{\mu}(1 - \epsilon/\|\tilde{\mu}\|_2) - q/2 < 0 \mid y = -1\right] \\
=&\tau\mathbb{P}\left[(\tilde{\mu} + w)^T\tilde{\mu}(1 - \epsilon/\|\tilde{\mu}\|_2) - q/2 > 0\right], \\
&+ (1-\tau)\mathbb{P}\left[(-\tilde{\mu} + w)^T\tilde{\mu}(1 - \epsilon/\|\tilde{\mu}\|_2) - q/2 < 0\right], \quad w \sim \mathcal{N}(0, I_d) \\
=&\tau\mathbb{P}\left[w^T\tilde{\mu}(1 - \epsilon/\|\tilde{\mu}\|_2) > q/2 - \tilde{\mu}^T\tilde{\mu}(1 - \epsilon/\|\tilde{\mu}\|_2)\right], \\
&+ (1-\tau)\mathbb{P}\left[w^T\tilde{\mu}(1 - \epsilon/\|\tilde{\mu}\|_2) < q/2 + \tilde{\mu}^T\tilde{\mu}(1 - \epsilon/\|\tilde{\mu}\|_2)\right], \quad w \sim \mathcal{N}(0, I_d) \\
=&\tau\mathbb{P}\left[\frac{w^T\tilde{\mu}(1 - \epsilon/\|\tilde{\mu}\|_2)}{\|\tilde{\mu}(1 - \epsilon/\|\tilde{\mu}\|_2)\|_2} > \frac{q/2 - \tilde{\mu}^T\tilde{\mu}(1 - \epsilon/\|\tilde{\mu}\|_2)}{\|\tilde{\mu}(1 - \epsilon/\|\tilde{\mu}\|_2)\|_2}\right], \\
&+ (1-\tau)\mathbb{P}\left[\frac{w^T\tilde{\mu}(1 - \epsilon/\|\tilde{\mu}\|_2)}{\|\tilde{\mu}(1 - \epsilon/\|\tilde{\mu}\|_2)\|_2} < \frac{q/2 + \tilde{\mu}^T\tilde{\mu}(1 - \epsilon/\|\tilde{\mu}\|_2)}{\|\tilde{\mu}(1 - \epsilon/\|\tilde{\mu}\|_2)\|_2}\right], \quad \frac{w^T\tilde{\mu}(1 - \epsilon/\|\tilde{\mu}\|_2)}{\|\tilde{\mu}(1 - \epsilon/\|\tilde{\mu}\|_2)\|_2} \sim \mathcal{N}(0, 1) \\
=&\tau\Phi(\frac{\tilde{\mu}^T\tilde{\mu}(1 - \epsilon/\|\tilde{\mu}\|_2) - q/2}{\|\tilde{\mu}(1 - \epsilon/\|\tilde{\mu}\|_2)\|_2}) + (1-\tau)\Phi(\frac{\tilde{\mu}^T\tilde{\mu}(1 - \epsilon/\|\tilde{\mu}\|_2) + q/2}{\|\tilde{\mu}(1 - \epsilon/\|\tilde{\mu}\|_2)\|_2}).
\end{aligned}$$

Let $w$ denote $\tilde{\mu}(1 - \epsilon/\|\tilde{\mu}\|_2)$, the we got the accuracy

$$a = \tau\Phi(\frac{\tilde{\mu}^T w - q/2}{\|w\|_2}) + (1-\tau)\Phi(\frac{\tilde{\mu}^T w + q/2}{\|w\|_2}).$$

(iv) For sample $x \sim P_{\mu_1, \mu_2, I_d}$, let $t$ denote $x - \frac{\mu_1 + \mu_2}{2}$, and $w$ denote $\tilde{\mu}(1 - \epsilon/\|\tilde{\mu}\|_2)$. According to Theorem 3(iii), when $\tilde{\mu}^T\tilde{\mu}(1 - \epsilon/\|\tilde{\mu}\|_2) - q/2 > 0$, the accuracy would be higher than 0.5. Therefore we consider $\tilde{\mu}^T w - q/2 > 0$.

Now consider the classifier in equation 5 and the corresponding scaled bound from (ii),

$$\|\bar{\Delta}\|_2 = \frac{|(x - \frac{\mu_1 + \mu_2}{2})^T\tilde{\mu}(1 - \epsilon/\|\tilde{\mu}\|_2) - q/2|}{|\tilde{\mu}^T\tilde{\mu}(1 - \epsilon/\|\tilde{\mu}\|_2) - q/2|} = \frac{|t^T w - q/2|}{|\tilde{\mu}^T w - q/2|} = \frac{|t^T w - q/2|}{\tilde{\mu}^T w - q/2}.$$

Since $t|y \sim \mathcal{N}(y\tilde{\mu}, I_d)$, we have $t^T w - q/2|y \sim \mathcal{N}(y\tilde{\mu}^T w - q/2, w^T w)$. When we only want to get the expected scaled bound of the correctly-classified samples, we have that

$$\begin{aligned}
\mathbb{E}\left[\|\bar{\Delta}\|_2 \mid f_\epsilon(x) = y\right] &= \frac{1}{\tilde{\mu}^T w - q/2}\mathbb{E}\left[|t^T w - q/2| \mid f_\epsilon(x) = y\right] \\
&= \frac{\tau}{\tilde{\mu}^T w - q/2}\mathbb{E}\left[|t^T w - q/2| \mid f_\epsilon(x) = y = 1\right] \\
&\quad + \frac{1 - \tau}{\tilde{\mu}^T w - q/2}\mathbb{E}\left[|t^T w - q/2| \mid f_\epsilon(x) = y = -1\right] \\
&= \frac{\tau}{\tilde{\mu}^T w - q/2}\mathbb{E}\left[t^T w - q/2 \mid y = 1, t^T w - q/2 \geq 0\right] \\
&\quad + \frac{1 - \tau}{\tilde{\mu}^T w - q/2}\mathbb{E}\left[-t^T w + q/2 \mid y = -1, t^T w - q/2 < 0\right].
\end{aligned}$$

Recall that $t^T w - q/2 | y \sim \mathcal{N}(y\tilde{\mu}^T w - q/2, w^T w)$, then by the mean of truncated normal distribution, it is true that

$$\mathbb{E}\left[t^T w - q/2 \mid y = 1, t^T w - q/2 \geq 0\right] = \tilde{\mu}^T w - q/2 + \|w\|_2 \frac{\phi(\frac{0 - \tilde{\mu}^T w + q/2}{\|w\|_2})}{1 - \Phi(\frac{0 - \tilde{\mu}^T w + q/2}{\|w\|_2})}$$

$$= \tilde{\mu}^T w - q/2 + \|w\|_2 \frac{\phi(\frac{-\tilde{\mu}^T w + q/2}{\|w\|_2})}{\Phi(\frac{\tilde{\mu}^T w - q/2}{\|w\|_2})}$$

$$\mathbb{E}\left[-t^T w + q/2 \mid y = -1, t^T w - q/2 < 0\right] = -\mathbb{E}\left[t^T w - q/2 \mid y = -1, t^T w - q/2 < 0\right]$$

$$= -\left(-\tilde{\mu}^T w - q/2 - \|w\|_2 \frac{\phi(\frac{0 + \tilde{\mu}^T w + q/2}{\|w\|_2})}{\Phi(\frac{0 + \tilde{\mu}^T w + q/2}{\|w\|_2})}\right)$$

$$= \tilde{\mu}^T w + q/2 + \|w\|_2 \frac{\phi(\frac{\tilde{\mu}^T w + q/2}{\|w\|_2})}{\Phi(\frac{\tilde{\mu}^T w + q/2}{\|w\|_2})}$$

Therefore

$$\mathbb{E}\left[\|\bar{\Delta}\|_2 \mid f_\epsilon(x) = y\right] = \frac{\tau}{\tilde{\mu}^T w - q/2}\left(\tilde{\mu}^T w - q/2 + \|w\|_2 \frac{\phi(\frac{-\tilde{\mu}^T w + q/2}{\|w\|_2})}{\Phi(\frac{\tilde{\mu}^T w - q/2}{\|w\|_2})}\right)$$

$$+ \frac{1 - \tau}{\tilde{\mu}^T w - q/2}\left(\tilde{\mu}^T w + q/2 + \|w\|_2 \frac{\phi(\frac{\tilde{\mu}^T w + q/2}{\|w\|_2})}{\Phi(\frac{\tilde{\mu}^T w + q/2}{\|w\|_2})}\right)$$

$\square$

# E ADDITIONAL TABLES

| $a_t$ | Model | $\epsilon = 0$ | $\epsilon = 0.2$ | $\epsilon = 0.4$ | $\epsilon = 0.6$ | $\epsilon = 0.8$ | CIFAR10 | CIFAR10-c | ImageNet | ImageNet-c |
|---|---|---|---|---|---|---|---|---|---|---|
| 0.7 | ViT-B/16 | 0.33 | 0.37 | 0.32 | 0.20 | 0.06 | 95.0 | 81.2 | 83.8 | 66.4 |
| | ViT-B/16-in21k | 0.20 | 0.23 | 0.18 | 0.07 | 0.01 | 89.6 | 71.4 | 82.6 | 63.6 |
| 0.75 | ViT-B/16 | 0.26 | 0.30 | 0.25 | 0.11 | 0.01 | 95.0 | 81.2 | 83.8 | 66.4 |
| | ViT-B/16-in21k | 0.12 | 0.16 | 0.10 | 0.02 | 0 | 89.6 | 71.4 | 82.6 | 63.6 |
| 0.8 | ViT-B/16 | 0.19 | 0.23 | 0.17 | 0.04 | 0 | 95.0 | 81.2 | 83.8 | 66.4 |
| | ViT-B/16-in21k | 0.06 | 0.09 | 0.04 | 0 | 0 | 89.6 | 71.4 | 82.6 | 63.6 |
| 0.85 | ViT-B/16 | 0.10 | 0.15 | 0.09 | 0 | 0 | 95.0 | 81.2 | 83.8 | 66.4 |
| | ViT-B/16-in21k | 0.01 | 0.02 | 0 | 0 | 0 | 89.6 | 71.4 | 82.6 | 63.6 |
| 0.9 | ViT-B/16 | 0.02 | 0.04 | 0.01 | 0 | 0 | 95.0 | 81.2 | 83.8 | 66.4 |
| | ViT-B/16-in21k | 0 | 0 | 0 | 0 | 0 | 89.6 | 71.4 | 82.6 | 63.6 |

Table 7: Full table of Table 2.

| $a_t$ | Model | $\epsilon = 0$ | $\epsilon = 0.2$ | $\epsilon = 0.4$ | $\epsilon = 0.6$ | $\epsilon = 0.8$ | CIFAR10 | CIFAR10-c | ImageNet | ImageNet-c |
|---|---|---|---|---|---|---|---|---|---|---|
| 0.7 | ViT-B/16 | 0.33 | 0.37 | 0.32 | 0.20 | 0.06 | 95.0 | 81.2 | 83.8 | 66.4 |
| | ViT-B/32 | 0 | 0 | 0 | 0 | 0 | 92.2 | 76.6 | 80.5 | 61.4 |
| 0.75 | ViT-B/16 | 0.26 | 0.30 | 0.25 | 0.11 | 0.01 | 95.0 | 81.2 | 83.8 | 66.4 |
| | ViT-B/32 | 0 | 0 | 0 | 0 | 0 | 92.2 | 76.6 | 80.5 | 61.4 |
| 0.8 | ViT-B/16 | 0.19 | 0.23 | 0.17 | 0.04 | 0 | 95.0 | 81.2 | 83.8 | 66.4 |
| | ViT-B/32 | 0 | 0 | 0 | 0 | 0 | 92.2 | 76.6 | 80.5 | 61.4 |
| 0.85 | ViT-B/16 | 0.10 | 0.15 | 0.09 | 0 | 0 | 95.0 | 81.2 | 83.8 | 66.4 |
| | ViT-B/32 | 0 | 0 | 0 | 0 | 0 | 92.2 | 76.6 | 80.5 | 61.4 |
| 0.9 | ViT-B/16 | 0.02 | 0.04 | 0.01 | 0 | 0 | 95.0 | 81.2 | 83.8 | 66.4 |
| | ViT-B/32 | 0 | 0 | 0 | 0 | 0 | 92.2 | 76.6 | 80.5 | 61.4 |

Table 8: Full table of Table 5.

| | arg max$_\epsilon$ SynBench-Score | linear probing | | 0.1-robust linear probing | | 0.2-robust linear probing | | 0.3-robust linear probing | | 0.4-robust linear probing | |
|---|---|---|---|---|---|---|---|---|---|---|---|
| | | CIFAR10 | CIFAR10-c | CIFAR10 | CIFAR10-c | CIFAR10 | CIFAR10-c | CIFAR10 | CIFAR10-c | CIFAR10 | CIFAR10-c |
| ViT-Ti/16 | 0.1 | 80.47±0.17 | 58.92±0.39 | 79.00±0.13 | 59.05±0.22 | 76.15±0.13 | 58.21±0.41 | - | - | - | - |
| ViT-B/16 | 0.2 | 94.47±0.05 | 80.65±0.15 | 95.17±0.04 | 81.55±0.33 | 95.09±0.05 | 81.99±0.18 | 95.01±0.05 | 81.87±0.43 | - | - |
| ViT-L/16 | 0.2 | 97.97±0.03 | 90.28±0.14 | 98.20±0.02 | 90.38±0.56 | 98.35±0.04 | 90.66±0.26 | 98.29±0.10 | 90.91±0.18 | - | - |
| ViT-B/16-in21k | 0.1/0.2 | 88.92±0.36 | 70.84±0.37 | 89.25±0.03 | 71.51±0.43 | 89.13±0.14 | 71.55±0.29 | 89.22±0.16 | 71.43±0.10 | - | - |
| ViT-B/32 | 0.2/0.3 | 92.02±0.15 | 75.75±0.33 | 92.48±0.04 | 76.92±0.30 | 92.44±0.06 | 76.92±0.27 | 92.41±0.07 | 77.14±0.18 | 92.43±0.05 | 76.96±0.05 |

Table 9: Full table of Table 6. '/' means the numbers are similar.

# F SYNTHETIC DATASET COMPLEXITY

SynBench can adjust the synthetic task complexity by statistically modeling the structure of the co-variance matrix $\mathbb{P}_\Sigma$. In our previous experiments, we have considered an identity covariance matrix, here we assume a channel-wise band matrix covariance $\Sigma$ (R,G,B channel entries are externally independent, and internally Gaussians with a band matrix covariance). Essentially, the assembled $\Sigma$ is a block-diagonal matrix with each block being a band matrix in the size of the image. For ViT with ImageNet size inputs, the 3 blocks are $224^2 \times 224^2$. We let the 3 band matrices be Pentadiagonal matrices (only main, first two upper, and two lower diagonals are nonzero).

From Table 10 to Table 13, we see that the trend suggested by SynBench is generally consistent with our findings in Section 4. Moreover, SynBench can well-capture the complexity of the task and give lower SynBench-scores when the task is more complex.

| $a_t = 0.7$ | $\epsilon = 0$ | $\epsilon = 0.2$ | $\epsilon = 0.4$ | $\epsilon = 0.6$ | $\epsilon = 0.8$ |
|---|---|---|---|---|---|
| ViT-B/16 | 0.18 | 0.24 | 0.20 | 0.10 | 0.01 |
| ViT-B/16-in21k | 0.07 | 0.11 | 0.07 | 0.01 | 0 |

Table 10: SynBench-Score comparisons on the finetuning procedure in pretraining on synthetic data with Pentadiagonal covariance.

| $a_t = 0.7$ | $\epsilon = 0$ | $\epsilon = 0.2$ | $\epsilon = 0.4$ | $\epsilon = 0.6$ | $\epsilon = 0.8$ |
|---|---|---|---|---|---|
| ViT-Ti/16 | 0 | 0 | 0 | 0 | 0 |
| ViT-B/16 | 0.18 | 0.24 | 0.20 | 0.10 | 0.01 |
| ViT-L/16 | 0.18 | 0.28 | 0.28 | 0.23 | 0.12 |

Table 11: SynBench-Score comparisons on the model sizes on synthetic data with Pentadiagonal covariance.

| $a_t = 0.7$ | $\epsilon = 0$ | $\epsilon = 0.2$ | $\epsilon = 0.4$ | $\epsilon = 0.6$ | $\epsilon = 0.8$ |
|---|---|---|---|---|---|
| ViT-S/16-DINO | 0.47 | 0.46 | 0.39 | 0.23 | 0.03 |
| ViT-B/16-DINO | 0.42 | 0.52 | 0.51 | 0.45 | 0.35 |
| ViT-S/8-DINO | 0.36 | 0.38 | 0.36 | 0.30 | 0.20 |
| ViT-B/8-DINO | 0.42 | 0.55 | 0.50 | 0.40 | 0.28 |
| Res50-SimCLRv2 | 0.24 | 0.47 | 0.36 | 0.33 | 0.31 |
| Res101-SimCLRv2 | 0.30 | 0.37 | 0.32 | 0.30 | 0.29 |

Table 12: SynBench-Scores of self-supervised pretrained representations on synthetic data with Pentadiagonal covariance.

| $a_t = 0.6$ | $\epsilon = 0$ | $\epsilon = 0.2$ | $\epsilon = 0.4$ | $\epsilon = 0.6$ | $\epsilon = 0.8$ |
|---|---|---|---|---|---|
| ViT-B/16 | 0.18 | 0.24 | 0.20 | 0.10 | 0.01 |
| ViT-B/32 | 0 | 0 | 0 | 0 | 0 |

Table 13: SynBench-Score comparisons on the ViT patch size of ViTs on synthetic data with Pentadiagonal covariance.

# G OTHER BASELINES

For completeness, we report several baseline metrics for the synthetic conditional Gaussian classification task. We follow the implementation of Whitney et al. (2020)[4] and set $ns$ (the training set size) to be the length of the synthetic dataset to compute canonical results. In Table 15, We report validation loss (val loss), minimum description length (MDL) (Voita & Titov, 2020), surplus description length (SDL) and $\epsilon$-sample complexity ($\epsilon$-SC) (Whitney et al., 2020). As a reference, obtaining the metrics for ViT-B/16 costs 6807 seconds and ViT-L/16 costs 7373 seconds on one Tesla V100.

---

[4]https://github.com/willwhitney/reprieve

| n | Name | ViT-B/16 | ViT-B/16-in21k |
|---|---|---|---|
| 2048 | Val loss | 3.10 | 3.37 |
| | MDL | 6820.76 | 7114.12 |
| | SDL, $\varepsilon$=1 | $> 4977.76$ | $> 5271.12$ |
| | $\varepsilon$SC, $\varepsilon$=1 | $> 1843.0$ | $> 1843.0$ |
| | SynBench | 0.33 | 0.20 |
| 4096 | Val loss | 1.77 | 1.41 |
| | MDL | 10813.95 | 9412.53 |
| | SDL, $\varepsilon$=1 | $> 7127.95$ | $> 5726.53$ |
| | $\varepsilon$SC, $\varepsilon$=1 | $> 3686.0$ | $> 3686.0$ |
| | SynBench | 0.45 | 0.30 |
| 8192 | Val loss | 0.73 | 0.77 |
| | MDL | 9939.13 | 9773.16 |
| | SDL, $\varepsilon$=1 | 3479.59 | 3153.33 |
| | $\varepsilon$SC, $\varepsilon$=1 | 7372 | 7372 |
| | SynBench | 0.52 | 0.38 |
| 16384 | Val loss | 0.85 | 0.86 |
| | MDL | 20936.18 | 20899.58 |
| | SDL, $\varepsilon$=1 | 7266.8 | 7136.29 |
| | $\varepsilon$SC, $\varepsilon$=1 | 14745 | 14745 |
| | SynBench | 0.56 | 0.41 |
| 32768 | Val loss | 0.68 | 0.70 |
| | MDL | 30848.99 | 32944.76 |
| | SDL, $\varepsilon$=1 | 7043.32 | 8611.49 |
| | $\varepsilon$SC, $\varepsilon$=1 | 14265 | 14265 |
| | SynBench | 0.59 | 0.44 |

Table 14: Baseline metrics evaluating the representation quality on the conditional Gaussian synthetic data with $n = \{2048, 4096, 8192, 16384, 32768\}$. For Val loss, MDL, SDL, and $\epsilon$SC, the smaller the better; for SynBench, the bigger the better. Note that the model ranking of SynBench is consistent across different values of $n$, while other methods will change their rankings.

| Name | ViT-Ti/16 | ViT-B/16 | ViT-L/16 | ViT-B/16-in21k | ViT-B/32 | ViT-S/16-DINO | ViT-S/8-DINO | ViT-B/16-DINO | ViT-B/8-DINO | Resnet50-SimCLRv2 | Resnet101-SimCLRv2 |
|---|---|---|---|---|---|---|---|---|---|---|---|
| Val loss | 4.38 | 0.73 | 1.50 | 0.77 | 2.92 | 1.51 | 0.70 | 0.92 | 0.64 | 0.62 | 0.52 |
| MDL | 30071.64 | 9939.13 | 17672.6 | 9773.16 | 23332.98 | 18536.93 | 8196.8 | 10535.11 | 6796.87 | 9646.09 | 5443.43 |
| SDL, $\varepsilon{=}1$ | $>22699.64$ | 3479.59 | $>10300.6$ | 3153.33 | $>15960.98$ | $>11164.93$ | 2056.69 | 3432.28 | 1185.31 | 3700.73 | 776.38 |
| $\varepsilon$SC, $\varepsilon{=}1$ | $>7372.0$ | 7372 | $>7372.0$ | 7372 | $>7372.0$ | $>7372.0$ | 4045 | 7372 | 2220 | 4045 | 669 |

Table 15: Baseline metrics evaluating the representation quality on the conditional Gaussian synthetic data with $n = 8192$.

