# OpenReview forum: "SynBench: Task-Agnostic Benchmarking of Pretrained Representations using Synthetic Data"
_ICLR.cc/2023/Conference — Submitted to ICLR 2023_

### Official Review · Reviewer_Lj97 · 2022-10-25

**Confidence:** 4
**Correctness:** 3
**Technical Novelty And Significance:** 2
**Empirical Novelty And Significance:** 2
**Recommendation:** 3

**Clarity, Quality, Novelty And Reproducibility:**

Over the paper is clear and easy to read.  Some of the captions could be slightly clearer.  While the reader can find alpha, epsilon in the text, it is helpful to restate that in the caption if possible.

**Strength And Weaknesses:**

Strengths:
Determining a robust and straightforward measure of the quality of learned representations (whether pretrained or otherwise) is an important endeavor.  Current approaches use linear-probing, but this is naturally task specific.  Having a task-agnostic measure of representation quality would be valuable.  The author's state the main tradeoff is between accuracy and adversarial robustness.

Weakness:
It's not clear whether a task-agnostic representation is meaningful or valuable.  Is the best representation the one that transfers to the most tasks, transfers the best to a specific task, is most extensible to an expanse of the domain, etc?  The authors have clearly stated what they deem as "good", but it's not clear that this is a general definition of a "good" representation.

The theoretical results derived all assume a balanced dataset, which is uncommon in real wold data.

The evaluation is only done on image classification (CIFAR 10/10-c).  While it is true that this dataset is used, it doesn't seem to be a good test dataset alone for this.  Vision transformers and large scale pretraining are most beneficial on massive datasets (both in the number of samples and when the images themselves are larger than CIFAR-10).  Therefore it's unclear how these results would hold on real world scenarios where such approaches would be actually used.  It's fine to start with CIFAR-10, but demonstration on a larger dataset (even Imagenet) and especially on a dataset where such approaches are more appropriate (e.g. medical imaging) would be much more powerful.

To demonstrate the usefulness of this approach, I believe the authors would need to more systematically demonstrate how the scores obtained from different pretraining procedure (e.g. MOCO, DINO, SimCLR, etc.), pretraining dataset (e.g. Imagenet, COCO, something entirely different like medical or remote sensing), downstream task (e.g. classification, segmentation, surface normals), and downstream dataset change and how that corresponds to the final downstream task performance (both fine tuning and linear probing).  The authors are implying that this metric is more general and therefore more useful than something like linear probing, but they have not demonstrated it over a wide enough range of representations and tasks, in my opinion.

**Summary Of The Paper:**

The authors propose a task-agnostic framework to measure the quality of pretrained representations.  This approach uses generated data from a conditional Gaussian mixture to evaluate the learned representation.  The usefulness of this metric is evaluated by comparing to results from linear probing on CIFAR 10/10-c

**Summary Of The Review:**

Creation of a task-agnostic measure of a learned pretrained representation has significant value.

However, by evaluating this metric only on a single task and dataset (classification for CIFAR 10/10-c), the impact of the paper is strongly limited.  How this would hold on other tasks (e.g. segmentation, surface normals, etc.) as well as larger, real world datasets where such approaches are needed, is not explored.  The assertion is that this metric is more useful than linear probing because it can be examined at the time of pretraining (not downstream task formulation) to anticipate how well that representation will perform across a number of tasks.  With that logic, this metric should enable the user to select the "best" representation from a set of representations which have been learned.  This needs to be demonstrated over a wide range of pretraining datasets, pretraining methodologies, downstream tasks, donwstream datasets, etc.

---

> ### Author Response · Authors · 2022-11-18
> **Authors' response (1/2)**
>
> ### 1. Defining good representations.
> The reviewer's comment is exactly the main motivation of this study.
> As it is generally hard to define whether a representation is good or bad in a task-agnostic and data-free way, we take the first step to narrow our scope for a more "well-posed" problem, by using a holistic  conditinal Gaussian mixture model to generate tractable synthetic tasks with known optimal robustness-accuracy performance as a reference to quantify the quality of representations. With the defined SynBench Score, we now can quantify the representations are better for the synthetic tasks if the score gets closer to 1 (meaning achieving the optimal robustness-accuracy tradeoff). We also believe our framework can be extended to incorporate other aspects for a more comprehensive assessment of representation quality, such as fairness.
>
>
> ### 2. Imbalanced dataset theoretical results.
> We have added the theory for the class-imbalanced class-conditional Gaussians with non-symmetric means and identify covariance case (Appendix D). Besides, we also generalize the current theorem to the general $\ell_p$ norm bound (Appendix C).

---

> ### Author Response · Authors · 2022-11-18
> **Authors' response (2/2)**
>
> ### 3. More experiments.
> In Section 4, we have added 6 more pretrained architectures, performed Imagenet/Imagenet-c linear probing (an example of the updates is given below and please see the updated manuscript, Table 2-5, for more results.), and included reference to other literature for task-specific results that requires access to the downstream data. Specifically, per reviewer's suggestions, we have conducted experiments on publicly available pretrained models from DINO and SimCLR, which are pretrained on Imagenet. With the added results, we observe that when we increase the model size, SynBench-score also increases: ViT-B/16-DINO, ViT-B/16-DINO, and Res101-SimCLRv2 have bigger SynBench-scores compared to ViT-S/16-DINO, ViT-S/16-DINO, and Res50-SimCLRv2, respectively.
>
> | $a_t=0.7$| $\epsilon=0$ | $\epsilon=0.2$ | $\epsilon=0.4$ | $\epsilon=0.6$ | $\epsilon=0.8$ | CIFAR10 | CIFAR10-c | ImageNet | ImageNet-c |
> | -------- | -------- | -------- | -------- | -------- | -------- | -------- | -------- | -------- | -------- |
> | ViT-S/16-DINO | 0.48 | 0.47 | 0.42 | 0.32 | 0.17 | 95.3 | 75.9 | 75.3 | 47.5 |
> | ViT-B/16-DINO | 0.55 | 0.58 | 0.53 | 0.46 | 0.35 | 96.5 | 78.9 | 76.4 | 52.1 |
> | ViT-S/8-DINO | 0.40 | 0.42 | 0.39 | 0.34 | 0.26 | 96.2 | 78.0 | 79.0 | 53.9 |
> | ViT-B/8-DINO | 0.50 | 0.56 | 0.50 | 0.40 | 0.30 | 97.0 | 80.6 | 79.5 | 53.7 |
> | Res50-SimCLRv2 | 0.66 | 0.50 | 0.50 | 0.48 | 0.48 | 95.0 | 80.1 | 77.5 | 47.4|
> | Res101-SimCLRv2 | 0.60 | 0.64 | 0.55 | 0.51 | 0.48 | 95.6 | 80.9 | 78.7 | 50.1 |
>
> Table: The SynBench-Score of self-supervised pretrained representations and the linear probing accuracy on CIFAR10/ImageNet and transfer accuracy on CIFAR10-c/ImageNet-c.
>
> | $a_t=0.7$ | $\epsilon=0$ | $\epsilon=0.2$ | $\epsilon=0.4$ | $\epsilon=0.6$ | $\epsilon=0.8$ | CIFAR10 | CIFAR10-c | ImageNet | ImageNet-c |
> | -------- | -------- | -------- | -------- | -------- | -------- | -------- | -------- | -------- | -------- |
> | ViT-B/16 | 0.33 | 0.37 | 0.32 | 0.20 | 0.06 | 95.0 | 81.2 | 83.8 | 66.4 |
> | ViT-B/16-in21k | 0.20 | 0.23 | 0.18 | 0.07 | 0.01 | 89.6 | 71.4 | 82.6 | 63.6 |
>
> Table: Comparisons on the finetuning procedure in pretraining with added ImageNet and ImageNet-c results. ViT-B/16-in21k and ViT-B/16 are both pretrained on ImageNet21k, where ViT-B/16 is further finetuned on ImageNet1k.
>
> While SynBench can provide robustness-accuracy evaluation for any given representation networks, we note that performing model pretraining is beyond the scope of this paper. That being said, we are more than happy to adding more results in the coming weeks if the reviewer are interested in seeing any publicly available models' performance.
>
> For models' performance on different downstream tasks (classification, segmentation) and datasets, we have sourced several literatures that include 27 linear probing classification tasks from [[Radford et al., 2021](http://proceedings.mlr.press/v139/radford21a/radford21a.pdf)], 4 chest x-ray image classification task from [[Okolo et al., 2022](https://www.sciencedirect.com/science/article/pii/S0169260722005223)], a brain magnetic resonance imaging tumor classification task from [[Tummala et al., 2022](https://www.mdpi.com/1718-7729/29/10/590)], a food segmentation task from [[Wu et al., 2021](https://dl.acm.org/doi/abs/10.1145/3474085.3475201)] and image retrieval, copy detection, video segmentation tasks from [[Caron et al., 2021](https://openaccess.thecvf.com/content/ICCV2021/papers/Caron_Emerging_Properties_in_Self-Supervised_Vision_Transformers_ICCV_2021_paper.pdf)].
>
> ### 4. Captions.
> We have rewrote some of the captions and are happy to keep revising if the reviewer finds the current ones still not clear enough.
>
>
> Radford, Alec, et al. "Learning transferable visual models from natural language supervision." International Conference on Machine Learning. PMLR, 2021.
>
> Okolo, Gabriel Iluebe, Stamos Katsigiannis, and Naeem Ramzan. "IEViT: An enhanced vision transformer architecture for chest X-ray image classification." Computer Methods and Programs in Biomedicine 226 (2022): 107141.
>
> Tummala, Sudhakar, et al. "Classification of Brain Tumor from Magnetic Resonance Imaging Using Vision Transformers Ensembling." Current Oncology 29.10 (2022): 7498-7511.
>
> Wu, Xiongwei, et al. "A large-scale benchmark for food image segmentation." Proceedings of the 29th ACM International Conference on Multimedia. 2021.
>
> Caron, Mathilde, et al. "Emerging properties in self-supervised vision transformers." Proceedings of the IEEE/CVF International Conference on Computer Vision. 2021.

---

### Official Review · Reviewer_xsCU · 2022-10-26

**Confidence:** 3
**Correctness:** 3
**Technical Novelty And Significance:** 3
**Empirical Novelty And Significance:** 2
**Recommendation:** 5

**Clarity, Quality, Novelty And Reproducibility:**

The paper is well-organized and easy to follow. The writing can be improved. For example, it is difficult to get an intuitive understanding of what area A and area B means without checking the equation details.

The idea is novel and interesting. But whether it is effective is uncertain given the result presented in the paper.

**Details Of Ethics Concerns:**

No ethics concerns.

**Strength And Weaknesses:**

Strength:
- The idea of evaluating a presentation without using a real dataset and downstream tasks is interesting. The idea is novel and the theory seems to be well proved under the Gaussian assumption.

Weakness:
- The theory's assumption is too strong and unclear whether the result can be transferred to more complex real-world scenarios.
- The experiment can be an effective way to validate whether the assumption holds true for more challenging scenarios. The experiment done in the paper is insufficient to get a conclusion. The comparison is done using variants of ViT and does not include other architectures. Since the evaluation method is a universal approach, it is important to see whether it is effective for other models.



**Summary Of The Paper:**

This paper presents a novel way to evaluate the quality of a pre-trained representation. It requires no real data and downstream tasks for the evaluation. In the evaluation, it utilizes synthetic data generated from the mixture of two Gaussian distributions. The authors prove the soundness of the method using the assumption of Gaussian.
The synthetic data is used to evaluate the trade-off between accuracy and robustness.
The evaluation is done on ViT architecture with different sizes. The experiment shows the SynBench score can be used to predict the model performance on downstream tasks.


**Summary Of The Review:**

The paper presents an interesting idea to evaluate pre-trained representation. The idea is conducted with Gaussian synthetic data. It is unclear to me whether this way of evaluation can really transfer to a meaningful metric for downstream tasks. The experiment in the paper is somewhat limited to support the claim of the authors. If authors can apply this evaluation strategy to more model architectures and show the same discovery, it will be very impressive. I think its current status is below the threshold of ICLR.

---

> ### Author Response · Authors · 2022-11-18
> **Authors' response (1/2)**
>
> ### 1. Use the theory to capture more complex real-world scenarios.
> Our theory holds for class-balanced class-conditional Gaussians with non-symmetric means and general covariance.
> Using this synthetic data framework, we can generate datasets of different complexity by feeding in different covariance structure. We have added this result to the updated manuscript (Appendix F), where we use a channel-wise band matrix covariance $\Sigma$ (R,G,B channel entries are externally independent, and internally Gaussians with a band matrix covariance).
>
> Additional theorical generalizations:
> * In the updated manuscript, we have also extended our theory to general $\ell_p$ norm bound (Appendix C), and
> * included the theory for the class-imbalanced class-conditional Gaussians with non-symmetric means and identify covariance case (Appendix D).
>
>
> ### 2. More experiments.
> In the updated manuscript, we have added these new experimental results:
> * In Section 4, we have added 6 more pretrained architectures that includes 4 self-supervied pretrained ViT [[Caron et al., 2021, DINO](https://openaccess.thecvf.com/content/ICCV2021/papers/Caron_Emerging_Properties_in_Self-Supervised_Vision_Transformers_ICCV_2021_paper.pdf)] and 2 ResNets [[Chen et al., 2020c, SimCLRv2](https://proceedings.neurips.cc/paper/2020/file/fcbc95ccdd551da181207c0c1400c655-Paper.pdf)]. This inclusion shows the universal use of SynBench in evaluating for different pretraining procedure, and different architectures. In fact, SynBench can provide robustness-accuracy evaluation for any given representations models. We are happy to continue adding more results in the coming weeks if the reviewer are interested in seeing any publicly available models' performance. With the added results, we observe that when we increase the model size, SynBench-score also increases: ViT-B/16-DINO, ViT-B/16-DINO, and Resnet101-SimCLRv2 have bigger SynBench-scores compared to ViT-S/16-DINO, ViT-S/16-DINO, and Resnet50-SimCLRv2, respectively.
>
> | $a_t=0.7$| $\epsilon=0$ | $\epsilon=0.2$ | $\epsilon=0.4$ | $\epsilon=0.6$ | $\epsilon=0.8$ | CIFAR10 | CIFAR10-c | ImageNet | ImageNet-c |
> | -------- | -------- | -------- | -------- | -------- | -------- | -------- | -------- | -------- | -------- |
> | ViT-S/16-DINO | 0.48 | 0.47 | 0.42 | 0.32 | 0.17 | 95.3 | 75.9 | 75.3 | 47.5 |
> | ViT-B/16-DINO | 0.55 | 0.58 | 0.53 | 0.46 | 0.35 | 96.5 | 78.9 | 76.4 | 52.1 |
> | ViT-S/8-DINO | 0.40 | 0.42 | 0.39 | 0.34 | 0.26 | 96.2 | 78.0 | 79.0 | 53.9 |
> | ViT-B/8-DINO | 0.50 | 0.56 | 0.50 | 0.40 | 0.30 | 97.0 | 80.6 | 79.5 | 53.7 |
> | Res50-SimCLRv2 | 0.66 | 0.50 | 0.50 | 0.48 | 0.48 | 95.0 | 80.1 | 77.5 | 47.4|
> | Res101-SimCLRv2 | 0.60 | 0.64 | 0.55 | 0.51 | 0.48 | 95.6 | 80.9 | 78.7 | 50.1 |
>
> Table: The SynBench-Score of self-supervised pretrained representations and the linear probing accuracy on CIFAR10/ImageNet and transfer accuracy on CIFAR10-c/ImageNet-c.
>
> * Per Reviewer Lj97's suggestion, we have added Imagenet linear probing results. We give an example of the updates below and please see the updated manuscript (Table 2-5) for more results. We refer the reviewers to literatures for more task-specific benchmark results ([Radford et al., 2021](http://proceedings.mlr.press/v139/radford21a/radford21a.pdf); [Okolo et al., 2022](https://www.sciencedirect.com/science/article/pii/S0169260722005223); [Tummala et al., 2022](https://www.mdpi.com/1718-7729/29/10/590); [Wu et al., 2021](https://dl.acm.org/doi/abs/10.1145/3474085.3475201); [Caron et al., 2021](https://openaccess.thecvf.com/content/ICCV2021/papers/Caron_Emerging_Properties_in_Self-Supervised_Vision_Transformers_ICCV_2021_paper.pdf));
>
> | $a_t=0.7$ | $\epsilon=0$ | $\epsilon=0.2$ | $\epsilon=0.4$ | $\epsilon=0.6$ | $\epsilon=0.8$ | CIFAR10 | CIFAR10-c | ImageNet | ImageNet-c |
> | -------- | -------- | -------- | -------- | -------- | -------- | -------- | -------- | -------- | -------- |
> | ViT-B/16 | 0.33 | 0.37 | 0.32 | 0.20 | 0.06 | 95.0 | 81.2 | 83.8 | 66.4 |
> | ViT-B/16-in21k | 0.20 | 0.23 | 0.18 | 0.07 | 0.01 | 89.6 | 71.4 | 82.6 | 63.6 |
>
> Table: Comparisons on the finetuning procedure in pretraining with added ImageNet and ImageNet-c results. ViT-B/16-in21k and ViT-B/16 are both pretrained on ImageNet21k, where ViT-B/16 is further finetuned on ImageNet1k.

---

> ### Author Response · Authors · 2022-11-18
> **Authors' response (2/2)**
>
> * Per Reviewer REYd's suggestion, we have modified and included 4 baseline evaluation metrics to Appendix G. Spefically, we report these baseline metrics for the synthetic conditional Gaussian classification task with sample size being $n=\{2048, 4096, 8192, 16384, 32768\}$. Note that the model ranking of SynBench is consistent across different values of $n$, while other methods will change their rankings.
>
> | n | Name | ViT-B/16 | ViT-B/16-in21k |
> | -------- | -------- | -------- | -------- |
> | 2048 | Val loss | 3.10 | 3.37 |
> ||  MDL | 6820.76 | 7114.12|
> ||  SDL, $\varepsilon$=1| $>$ 4977.76 | $>$ 5271.12 |
> ||  $\varepsilon$SC, $\varepsilon$=1 | $>$ 1843.0 | $>$ 1843.0 |
> ||  SynBench | 0.33 | 0.20 |
> | 4096 | Val loss | 1.77 | 1.41 |
> ||  MDL | 10813.95 | 9412.53 |
> ||  SDL, $\varepsilon$=1 | $>$ 7127.95 | $>$ 5726.53 |
> ||  $\varepsilon$SC, $\varepsilon$=1  | $>$ 3686.0 | $>$ 3686.0 |
> ||  SynBench | 0.45 | 0.30 |
> | 8192 | Val loss | 0.73 | 0.77 |
> ||  MDL | 9939.13  | 9773.16 |
> ||  SDL, $\varepsilon$=1 | 3479.59 | 3153.33 |
> ||  $\varepsilon$SC, $\varepsilon$=1  &| 7372 | 7372 |
> ||  SynBench | 0.52 | 0.38 |
> | 16384 | Val loss | 0.85 | 0.86 |
> ||  MDL | 20936.18 | 20899.58 |
> ||  SDL, $\varepsilon$=1 | 7266.8 | 7136.29 |
> ||  $\varepsilon$SC, $\varepsilon$=1  | 14745 | 14745 |
> ||  SynBench | 0.56 | 0.41 |
> | 32768 | Val loss | 0.68 | 0.70 |
> ||  MDL | 30848.99 | 32944.76 |
> ||  SDL, $\varepsilon$=1 | 7043.32 | 8611.49 |
> ||  $\varepsilon$SC, $\varepsilon$=1  | 14265 | 14265 |
> ||  SynBench | 0.59 | 0.44 |
>
> Table: Baseline metrics evaluating the representation quality on the conditional Gaussian synthetic data with $n=\{2048, 4096, 8192, 16384, 32768\}$. For Val loss, MDL, SDL, and $\epsilon$SC, the smaller the better; for SynBench, the bigger the better. Note that the model ranking of SynBench is consistent across different values of $n$, while other methods will change their rankings. ''$>$'' means the evaluation dataset is insufficient to provide concrete quantities.
>
>
> ### 3. Writing.
> We have rewrote many parts of the paper (colored in red). We hope the reviewer will find it easier to get intuitive understandings now.
>
>
> Caron, Mathilde, et al. "Emerging properties in self-supervised vision transformers." Proceedings of the IEEE/CVF International Conference on Computer Vision. 2021.
>
> Chen, Ting, et al. "Big self-supervised models are strong semi-supervised learners." Advances in neural information processing systems 33 (2020): 22243-22255.
>
> Radford, Alec, et al. "Learning transferable visual models from natural language supervision." International Conference on Machine Learning. PMLR, 2021.
>
> Okolo, Gabriel Iluebe, Stamos Katsigiannis, and Naeem Ramzan. "IEViT: An enhanced vision transformer architecture for chest X-ray image classification." Computer Methods and Programs in Biomedicine 226 (2022): 107141.
>
> Tummala, Sudhakar, et al. "Classification of Brain Tumor from Magnetic Resonance Imaging Using Vision Transformers Ensembling." Current Oncology 29.10 (2022): 7498-7511.
>
> Wu, Xiongwei, et al. "A large-scale benchmark for food image segmentation." Proceedings of the 29th ACM International Conference on Multimedia. 2021.

---

### Official Review · Reviewer_REYd · 2022-11-02

**Confidence:** 3
**Correctness:** 2
**Technical Novelty And Significance:** 2
**Empirical Novelty And Significance:** 2
**Recommendation:** 3

**Clarity, Quality, Novelty And Reproducibility:**

### Clarity

Below, I've tried to explicit the points that were confusing to me.

*Abstract*. I would suggest removing some technical details and focus on the important message. In particular, the sentence "we set up a reference ... to infer the quality" is quite unclear. On one hand, introducing the use of gaussian mixtures feels unecessary at this stage. On the other hand, focusing on explicting what "accuracy-robustness" tradeoff means for a score that claims to be task-agnostic would be more insightful. Regarding that point, I'd suggest authors state explicitly that they create a synthetic binary classification proxy task to estimate the model's accuracy-robustness (hard to understand before page 4 of the current manuscript).

*Gap in argumentation.* Introduction starts by motivating the surge in usage of large pretrained models, and the need for a way to assess the quality of representations in a "task-agnostic" fashion, which I completly hear. With almost no transition, the argument jumps to a discussion on how a potential lack adversarial robustness from the network could impede the model's performance on downstream tasks. I had a hard time following the rest of the introduction from that point on, i.e. from "however, if the underlying ... standard accuracy and adversarial robustness to input perturbations...". I may be missing context, but I still suggest authors elaborate/provide references to bridge the gap between a model's adversarial robustness and its potential of better transfering to new tasks.

*Mathematical notations.*: I found mathematical expressions unadequately verbose.
  My high-level suggestions would be to defer as much as possible to Appendix (including at least 2 out of 4 lines of Eq (2)), and revise/simplify notations. Here are more specific points:

- I guessed by elimination that $y^*$ corresponded to the optimal $\epsilon$-robust classifier. However, classifiers were referred to as $f$
       in the section before. If my understanding is correct, I would suggest something like $f_{\epsilon}^{*}$.
- Unless I missed it: $\theta$ is not defined. I tried to assume that represented the network's parameters, but still does not make sense, since at the stage this is introduced, we're still referring the input space (in other words, there is no dependence upon the networks at all). Please clarify this.
- Adding $\mu=..., \sigma=...$ at each line makes Eq (2) unecesarrily bulky. Unless I'm missing something, the dependency upon $s$ would be better carried by $a$, such as $a(s_i, \epsilon) > a_t$. That would also allow $\epsilon$ to also appear as a dependency and ease reading.
- Results 3.1 to 3.4: After re-reading those 4 results, it appears to me there is a lot of repetition which could presumably be factorized into a single result giving both the expected $||\delta||$ and standard accuracy in the case of a general gaussian mixture, and subsequently instantiated with more assumptions (e.g. $\Sigma=\sigma I$).



**Strength And Weaknesses:**

### Strengths

**Interesting problem.** The problem of evaluating the quality of a pretrained model's representations in a *task-agnostic* fashion appears as an ambitious (perhaps a bit ill-posed?)/important problem in a context where pretrained/foundations models are becoming de-facto starting points for most existing specific applications.

**Exhaustive related work section.** The related works part covers relevant lines of works, including most recent works on pretrained models benchmarking.

### Weaknesses

**Motivation.**

*Use of synthetic data.* I'm having hard time wrapping my head around how the model's behavior on images synthetized from a Gaussian mixture (i.e probably very far away in the input space from both the source and target distributions) could inform the model's behavior on real-world images. Could authors provide additional motivation ?

  *Adversarial perspective: * (See below for a more detailed comment) I'd suggest authors provide more motivation on why they chose to address the general problem of "representation quality" through the lens of adversarial robustness, as the link between adversarial robustness and generalization is not trivial (although I concede it is not unreasonable either).

**Lack of comparisons.** Authors do not really compare to any other line of work. Although I could concede that other works have never tried to address this exact setting (i.e completly task-agnostic evaluation of representations), authors do not seem to have put any effort into adapting any technique to their setting, which makes the relevance of their score (on top of my next comment) hard to evaluate. For instance, cited lines of work that use unsupervised criterions (e.g. mutual information or Minimum Description Length) could be adaptated to the synthetic binary classification task that authors propose (e.g. comparing the mutual information between representations and labels in the input space vs in the representation space), and provide at least some point of comparison.

**Hardly falsifiable hypothesis.** The contribution section claims that the score evaluates the quality of the representations. On the other hand, authors do not find any strong positive correlation between their score and actual tradeoff performances of models in Table 3. (e.g. standard probing for ViT-L, corresponding to $\epsilon=0$, has smaller score but significantly higher OOD generalization). Authors comment on that part that "although SyncBench-score may share trends with empricial real-life tasks, it is meant to characterize a general behavior of the pretrained representations." This feel a bit hand-wavy to me, in the sense that the definition of "general behavior" can mean everything and anything. I believe the paper would benefit from clearly defining what the score is meant to act as a proxy of, and explictly show positive correlation between their score and this metric.

**Lack of experiments.** Echoing to my previous point, even if strongly positive correlation had actually been found, more experiments would be needed to support the initial goal/claims of producing a score that *applies to a wide range of pretrained models* and informs downstream accuracy. In particular, more architectures should be tried (at least 1 ConvNet), more datasets (CIFAR is a relatively outdated choice considering the myriad of more recent vision datasets), and at least one additional task (e.g. segmentation/detection ? or even anything in another modality?) to support the *task-agnosticity* part.

**Summary Of The Paper:**

This paper aims at finding a *task-agnostic* and *model-agnostic* way of evaluating the quality of any pretrained model's extracted features. Specifically, authors develop a score that measures the accuracy-robustness tradeoff of a model's representation on a synthetic binary classification task. Authors asses the relevance of such score by studying how well it correlates with ViT model's ability to obtain a good in-distribution (CIFAR) / out-of-distribution (CIFAR-10C) accuracy tradeoff.

**Summary Of The Review:**

The current manuscript does not appear ready for publication due to (i) lacking motivation on the important directions of the work, (ii) lacking experiments/evidence to properly support author's claims and (iii) lacking clarity and conciness in writing. Therefore, I cannot recommend acceptance at this stage.

---

> ### Author Response · Authors · 2022-11-18
> **Authors' response (1/3)**
>
> ### 1. Motivation for the use of synthetic data.
> SynBench is an evaluation that serves as a sanity check for pretrained models when there is no access nor knowledge of the downstream task. We choose Gaussian mixtures for evaluating pre-trained models on images because it is shown to be capable of modeling the statistics of natural images  [[Zoran, Daniel, and Yair Weiss](https://proceedings.neurips.cc/paper/2012/file/e97ee2054defb209c35fe4dc94599061-Paper.pdf)]. On the other hand, we view SynBench as a ''necessary'' and ''minimum'' test in the sense that, with perfect (no data bias) data, any undesirable deteriorated behavior (such as weakened robustness) reveals the weaknesses the representation model. Therefore, in designing this minimum test, it is important that the task has a theoretical ideal (and optimal) solution (i.e. the trade-off preserved by class conditional Gaussians, Theorem 1 iv). We have updated the paper to give a better motivation.
>
> ### 2. Motivation for adversarial perspective and the gap in argumentation.
>
> It is known that standard accuracy does not correlate well (even worse, sometimes has negative correlation) with adversarial robustness [[Su, Dong, et al, 2018]](https://openaccess.thecvf.com/content_ECCV_2018/papers/Dong_Su_Is_Robustness_the_ECCV_2018_paper.pdf). Therefore, for practicality and utility, a good pre-trained model should have both good accuracy and adversarial robustness. Our proposed SynBech Score provides a quantifiable metric to evaluate such tradeoff between robustness and accuracy, and as a result, it can be used for performance bechmarcking and facilitates hyperparameter selection in linear probing, as shown in this paper.
>
> In defining good representations, we seek for both accuracy and robustness, in a task-agnostic sense. We have revised the introduction to reflect this point and bridged the gap in our argumentation.
>
> ### 3. Comparisons with prior arts.
> In the updated manuscript, Appendix G, we have added comparisons with prior arts that includes minimum description length (MDL) [[Elena Voita \& Ivan Titov, 2020](https://aclanthology.org/2020.emnlp-main.14.pdf)], surplus description length (SDL) and $\epsilon$-sample complexity ($\epsilon$-SC) [[Whitney et al., (2020](https://arxiv.org/pdf/2009.07368.pdf)]. Specifically, we report baseline metrics evaluating the representation quality on the conditional Gaussian synthetic data with sample size being $n=\{2048, 4096, 8192, 16384, 32768\}$. Note that the model ranking of SynBench is consistent across different values of $n$, while other methods will change their rankings. We did not include mutual information based modifications since estimating mutual information is itself intractable and a long-standing research problem [[McAllester \& Stratos, 2020](http://proceedings.mlr.press/v108/mcallester20a/mcallester20a.pdf)]. We also do not observe components in these baselines that would account for representational robustness.
>
> | n | Name | ViT-B/16 | ViT-B/16-in21k |
> | -------- | -------- | -------- | -------- |
> | 2048 | Val loss | 3.10 | 3.37 |
> ||  MDL | 6820.76 | 7114.12|
> ||  SDL, $\varepsilon$=1| $>$ 4977.76 | $>$ 5271.12 |
> ||  $\varepsilon$SC, $\varepsilon$=1 | $>$ 1843.0 | $>$ 1843.0 |
> ||  SynBench | 0.33 | 0.20 |
> | 4096 | Val loss | 1.77 | 1.41 |
> ||  MDL | 10813.95 | 9412.53 |
> ||  SDL, $\varepsilon$=1 | $>$ 7127.95 | $>$ 5726.53 |
> ||  $\varepsilon$SC, $\varepsilon$=1  | $>$ 3686.0 | $>$ 3686.0 |
> ||  SynBench | 0.45 | 0.30 |
> | 8192 | Val loss | 0.73 | 0.77 |
> ||  MDL | 9939.13  | 9773.16 |
> ||  SDL, $\varepsilon$=1 | 3479.59 | 3153.33 |
> ||  $\varepsilon$SC, $\varepsilon$=1  &| 7372 | 7372 |
> ||  SynBench | 0.52 | 0.38 |
> | 16384 | Val loss | 0.85 | 0.86 |
> ||  MDL | 20936.18 | 20899.58 |
> ||  SDL, $\varepsilon$=1 | 7266.8 | 7136.29 |
> ||  $\varepsilon$SC, $\varepsilon$=1  | 14745 | 14745 |
> ||  SynBench | 0.56 | 0.41 |
> | 32768 | Val loss | 0.68 | 0.70 |
> ||  MDL | 30848.99 | 32944.76 |
> ||  SDL, $\varepsilon$=1 | 7043.32 | 8611.49 |
> ||  $\varepsilon$SC, $\varepsilon$=1  | 14265 | 14265 |
> ||  SynBench | 0.59 | 0.44 |
>
> Table: Baseline metrics evaluating the representation quality on the conditional Gaussian synthetic data with $n=\{2048, 4096, 8192, 16384, 32768\}$. For Val loss, MDL, SDL, and $\epsilon$SC, the smaller the better; for SynBench, the bigger the better. Note that the model ranking of SynBench is consistent across different values of $n$, while other methods will change their rankings. ''$>$'' means the evaluation dataset is insufficient to provide concrete quantities.

---

> > ### Comment · Reviewer_REYd · 2022-11-22
> > **Post-rebuttal acknowledgment**
> >
> > I thank the reviewers for their great work during the rebuttal period.
> >
> > I feel the paper has been improved during the rebuttal, especially regarding the clarity and conciseness of the presentation (mathematical notations/presentation of the main result). I also appreciate that the authors have added more results and compared their method to new baselines. However, the authors have not fully addressed two important concerns, which are (a) the motivation for the adversarial perspective and (b) the correlation of their metric with real tasks.
> >
> > For (a), the authors state "*It is known that standard accuracy does not correlate well (even worse, sometimes has negative correlation) with adversarial robustness*"'. I agree with that, but this does necessarily hold true for natural (or even non-adversarial synthetic ones) distribution shifts (quite the opposite actually: in-distribution accuracy often correlates with OOD accuracy) that are used to empirically validate the method in the paper's experiments. Therefore, the motivation of the whole adversarial perspective remains blurry to me.
> >
> > (b) The reviewer's answer sounds like an impossibility result to me: if no single pretrained network can outperform other networks in every domain, how could the problem of a task-agnostic *comparison* of models be well defined? I believe this question should be addressed carefully in future revisions.
> >
> > Because I believe (a) and (b) are two central pieces of the work, I have decided to keep my initial score.

---

> > > ### Author Response · Authors · 2022-11-23
> > > **Address remaining concerns**
> > >
> > > ### Dear Reviewer  REYd,
> > > We are delignted to learn from your comment that "I feel the paper has been improved during the rebuttal, especially regarding the clarity and conciseness of the presentation (mathematical notations/presentation of the main result)." Based on your feedback, we now better understand your remaining two important concerns and will address them in this reponse.
> > >
> > > #### a. the motivation for the adversarial perspective
> > > For (a), we would like to clarify that our statement is not to be interpredted as dismissive of the studies on OOD accuracy. In fact, we hope the reviewer would agree that adversarial robustness (worst-case robustness) and OOD accuracy have fundamental differences. As the reviewer already pointed out,  non-adversarial synthetic data relate to OOD accuracy, while adversarial robustness handles adversarial examples. These two metrics (ODD accuracy and accuracy against adversairal examples) also may not be strongly correlated. For example, recent studies such as [R1] show that adversarial robustness does not transfer well from pre-trained models to fine-tuned tasks. Lacking adversarial robustness in downstream tasks could bring about many undesirable negative consequences to safety and security.
> > >
> > > In our paper and theroerical analysis, we have clearly stated that adversarial robustness refers to model accuracy in the worst-case scenario, and our robustness-accuracy plot is built upon the relationship between standard accuracy and the average margin to the closest decision boundary (as a meature of adversarial robustness). The motivation of our work lies in evaluating the robustness (against adversarial examples) and accuracy of pre-trained models in a task-agnostic and data-free manner. When using SynBench to select a better hyperparameter $\epsilon$ for adversarailly robust linear probing (such as Table 6), we also observe that it leads to better OOD accuracy on CIFAR-C/ImageNet-C. Therefore, we believe our SunBench framework is well motivated, as it provides new insights in evaluating the quality of pre-trained models and informs the design of robust linear probing methods.
> > >
> > > #### b. the correlation of their metric with real tasks
> > > Following the reviewer's suggestion, we conducted the evaluations for pretrained representations using SynBench and baselines we adapted in the Stage 1 rebuttal. We compare the ranking with the averaged accuracy from the Table 10 of [Radford et al., 2021]. We report the five pretrained models out of the overall eleven due to the lack of reported results from the literature for the other six pretrain models. Since ''$>$'' means the evaluation dataset is insufficient to provide concrete quantities, we treat the pretrained representations with ''$>$'' as tied. We make the comparisons using Kendall's Tau as the rank correlation metric.
> > >
> > > As can be seen from the table, SynBench provides the highest ranking correlation among the baselines when the size of the test set n is 2048. While MDL is able to give the same score as SynBench, it is sensitive to n as well as had been shown in our Stage 1 rebuttal point \#3. Baseline methods will catch up with SynBench as n grows large, but do not surpass SynBench.
> > >
> > >
> > >
> > >
> > >
> > > |n | Name | ViT-B/16 | ViT-L/16 | ViT-B/32 | Resnet50-SimCLRv2 | Resnet101-SimCLRv2  | ranking correlation|
> > > -------- | -------- | -------- | -------- | -------- |-------- | -------- | -------- |
> > > | | Real-life tasks acc. | 74.3 (4) | 75.5 (1) | 72.6 (5) | 75.4 (2) | 75.4 (2) | 1.0|
> > > |2048 | Val loss | 3.10 (3) | 4.12 (5) | 4.10 (4) | 1.31 (2) | 0.98 (1) | 0.11|
> > > | | MDL | 6820.76 (3) | 8094.06 (4) | 8198.55 (5) | 5881.34 (2) | 2882.36 (1) | 0.32|
> > > | | SDL, $\varepsilon$=1 | $>$ 4977.76 (5) | $>$ 6251.06 (5) | $>$ 6355.55 (5) | $>$ 4038.34 (5) | 1052.37 (1) | 0.17* |
> > > | | $\varepsilon$SC, $\varepsilon$=1 | $>$ 1843.0 (5) | $>$ 1843.0 (5) | $>$ 1843.0 (5) | $>$ 1843.0 (5) | 1843 (1) | 0.17*|
> > > | | SynBench | 0.33 (3) | 0.26 (4) | 0.02 (5) | 0.66 (1) | 0.60 (2) | 0.32|
> > > | 8192 | Val loss | 0.73 (3) | 1.50 (4) | 2.92 (5) | 0.62 (2) | 0.52 (1) | 0.32 |
> > > | | MDL | 9939.13 (3) | 17672.6 (4) | 23332.98 (5) | 9646.09 (2) | 5443.43 (1) | 0.32 |
> > > | | SDL, $\varepsilon$=1 | 3479.59 (2) | $>$ 10300.6 (5) | $>$ 15960.98 (5) | 3700.73 (3) | 776.38 (1) | 0* |
> > > | | $\varepsilon$SC, $\varepsilon$=1 | 7372 (3) | $>$ 7372.0 (5) | $>$ 7372.0 (5) | 4045 (2) | 669 (1) | 0.22*|
> > > | | SynBench | 0.52 (3) | 0.49 (4) | 0.01 (5) | 0.69 (2) | 0.84 (1) | 0.32|
> > > | 32768 | Val loss | 0.68 (3) | 0.79 (4) | 3.91 (5) | 0.53 (2) | 0.51 (1) | 0.32|
> > > | | MDL | 30848.99 (3) | 38718.04 (4) | 107960.49 (5) | 22022.08 (2) | 17166.0 (1) | 0.32|
> > > | | SDL, $\varepsilon$=1 | 7043.32 (3) | 12496.0 (4) | $>$ 78469.49 (5) | 4355.67 (2) | 969.27 (1) | 0.32*|
> > > | | $\varepsilon$SC, $\varepsilon$=1 | 14265 (3) | 29491 (4) | $>$ 29491.0 (5) | 3338 (2) | 1615 (1) | 0.32*|
> > > | | SynBench | 0.59 (3) | 0.58 (4) | 0.02 (5) | 0.81 (2) | 0.87 (1) | 0.32|
> > >
> > > Table: Campare the rankings of models by Kendall's tau.

---

> > > > ### Comment · Reviewer_REYd · 2022-12-12
> > > > **Answer to authors**
> > > >
> > > > I thank the authors for their reply and the new results.
> > > >
> > > > 1. Motivation of the work. I'm still having trouble wrapping my head around the sentence *The motivation of our work lies in evaluating the robustness (against adversarial examples) and accuracy of pre-trained models in a task-agnostic and data-free manner.*. In-fine, the experimental section largely relies on the OOD accuracy as a measure of the score's usefulness (for instance no evaluation of actual real-life adversarial accuracy). If authors wanted to make a strong case for using the score to choose the optimal $\epsilon$ (again, need to define *optimal* for what, except for the score itself), that would need to be done more rigorously. As it stands, we can only say the method is able to find some $\epsilon$ (that happens to be different from 0) and therefore quite unsurprisingly yields a "more robust" classifier.
> > > >
> > > > 2. This way of presenting results is interesting. But indeed it does confirm that the correlation remains relatively low (and this for all methods), which also questions whether informing real-life behavior of models from a task-agnostic / data-free evaluation is even doable.
> > > >
> > > > As a final summary, I still think the topic is interesting, but I think the angle of attack chosen by the paper is neither natural nor very well stated (or at least there remains a discrepancy between the main text's motivation and experiments). For these reasons, I'm keeping my original score.

---

> > > > > ### Author Response · Authors · 2022-12-12
> > > > > **Thank you for your constructive feedback**
> > > > >
> > > > > We would like to thank the reviewer for your constructive feedback.
> > > > >
> > > > > To address your first point, we will add AutoAttack results to provide more adversarial example viewpoints.
> > > > > To address your second point, we have further calculated the Pearson correlation coefficients: with 2k synthetic samples, SynBench gives 0.79 whereas Val loss and MDL give 0.55 and 0.50; with 8k synthetic samples, SynBench gives 0.89 whereas Val loss and MDL give 0.81 and 0.68. We will add these results to our future version.
> > > > >
> > > > >
> > > > >
> > > > > Thanks,
> > > > > Authors

---

> ### Author Response · Authors · 2022-11-18
> **Authors' response (2/3)**
>
> ### 4. Correlations with real-life tasks
> We fully understand the reviewer's concern but we believe the implications/conclusions suggested by any ''better'' results on specific datasets are subjective (to the datasets used for evaluation) and inconclusive (when the evaluation datasets change). For example, ViT-L/16 is reportedly performing worse than ViT-B/16 on 4 out of 27 linear probing tasks according to Table 10 of [[Radford et al., 2021](http://proceedings.mlr.press/v139/radford21a/radford21a.pdf)], and also lose to ViT-B/16 on finetuned medical tasks with, e.g., X-ray images [[Okolo et al., 2022, Table 4-8](https://www.sciencedirect.com/science/article/pii/S0169260722005223)], and magnetic resonance imaging [[Tummala et al., 2022, Table 2-3](https://www.mdpi.com/1718-7729/29/10/590)]. This is also the reason why we feel the necessity to develop task-agnostic and real-life data-free benchmarking methods.
>
>
> ### 5. More experiments.
> In the updated manuscript, Section 4, we have added 6 more pretrained architectures that includes 4 self-supervied pretrained ViT [[Caron et al., 2021, DINO](https://openaccess.thecvf.com/content/ICCV2021/papers/Caron_Emerging_Properties_in_Self-Supervised_Vision_Transformers_ICCV_2021_paper.pdf)] and 2 ResNets [[Chen et al., 2020c, SimCLRv2](https://proceedings.neurips.cc/paper/2020/file/fcbc95ccdd551da181207c0c1400c655-Paper.pdf)].
> | $a_t=0.7$| $\epsilon=0$ | $\epsilon=0.2$ | $\epsilon=0.4$ | $\epsilon=0.6$ | $\epsilon=0.8$ | CIFAR10 | CIFAR10-c | ImageNet | ImageNet-c |
> | -------- | -------- | -------- | -------- | -------- | -------- | -------- | -------- | -------- | -------- |
> | ViT-S/16-DINO | 0.48 | 0.47 | 0.42 | 0.32 | 0.17 | 95.3 | 75.9 | 75.3 | 47.5 |
> | ViT-B/16-DINO | 0.55 | 0.58 | 0.53 | 0.46 | 0.35 | 96.5 | 78.9 | 76.4 | 52.1 |
> | ViT-S/8-DINO | 0.40 | 0.42 | 0.39 | 0.34 | 0.26 | 96.2 | 78.0 | 79.0 | 53.9 |
> | ViT-B/8-DINO | 0.50 | 0.56 | 0.50 | 0.40 | 0.30 | 97.0 | 80.6 | 79.5 | 53.7 |
> | Res50-SimCLRv2 | 0.66 | 0.50 | 0.50 | 0.48 | 0.48 | 95.0 | 80.1 | 77.5 | 47.4|
> | Res101-SimCLRv2 | 0.60 | 0.64 | 0.55 | 0.51 | 0.48 | 95.6 | 80.9 | 78.7 | 50.1 |
>
> Table: The SynBench-Score of self-supervised pretrained representations and the linear probing accuracy on CIFAR10/ImageNet and transfer accuracy on CIFAR10-c/ImageNet-c.
>
> In terms of real-life task references, we have performed ImageNet/ImageNet-c linear probing and give an example of the updates below. Please see the updated manuscript (Table 2-5) for more results. Additionally, we referred the reviewers to the Table 10 of [[Radford et al., 2021](http://proceedings.mlr.press/v139/radford21a/radford21a.pdf)] for a good list of linear probing results. We also noticed some reported segmentation and detection results from Table 8 of [[Wu et al., 2021](https://dl.acm.org/doi/abs/10.1145/3474085.3475201)] and [[Caron et al., 2021, DINO](https://openaccess.thecvf.com/content/ICCV2021/papers/Caron_Emerging_Properties_in_Self-Supervised_Vision_Transformers_ICCV_2021_paper.pdf)].
>
>
>
> | $a_t=0.7$ | $\epsilon=0$ | $\epsilon=0.2$ | $\epsilon=0.4$ | $\epsilon=0.6$ | $\epsilon=0.8$ | CIFAR10 | CIFAR10-c | ImageNet | ImageNet-c |
> | -------- | -------- | -------- | -------- | -------- | -------- | -------- | -------- | -------- | -------- |
> | ViT-B/16 | 0.33 | 0.37 | 0.32 | 0.20 | 0.06 | 95.0 | 81.2 | 83.8 | 66.4 |
> | ViT-B/16-in21k | 0.20 | 0.23 | 0.18 | 0.07 | 0.01 | 89.6 | 71.4 | 82.6 | 63.6 |
>
> Table: Comparisons on the finetuning procedure in pretraining with added ImageNet and ImageNet-c results. ViT-B/16-in21k and ViT-B/16 are both pretrained on ImageNet21k, where ViT-B/16 is further finetuned on ImageNet1k.
>
>
> ### 6. Abstract.
> We made changes to the abstract according to Reviewer REYd's suggestion and stated clearly that we are using synthetic tasks to probe model's performance, in the sense of robustness and accuracy.

---

> ### Author Response · Authors · 2022-11-18
> **Authors' response (3/3)**
>
> ### 7. Mathematical notations.
> We have made all the changes suggested by the reviewer. Specifically, we
> 1. cut out 3 out of 4 lines of Eq (2) in the main text;
> 2. replaced $y^*$ by $f_\epsilon$;
> 3. defined $\theta$. To clarify, we have $\theta$ in the subscript because $E_{\theta,\epsilon}(a_{t})$ is a unified notation for both inputs and representations. We have clarified in Section 3.3, the third paragraph, that due to no dependency on the network parameter $\theta$, $E_{\theta,\epsilon}(a_{t})$ for the input data will be written as $E(a_{t})$, to distinguish from that for representations;
> 4. let $a$ carry the dependencies;
> 5. grouped Result 3.1 to 3.4 into Theorem 1, and referred to Theorem 1 results when we explain SynBench workflow in Section 3.2.
>
> Zoran, Daniel, and Yair Weiss. "Natural images, Gaussian mixtures and dead leaves." Advances in Neural Information Processing Systems 25. (2012).
>
> Su, Dong, et al. "Is Robustness the Cost of Accuracy?--A Comprehensive Study on the Robustness of 18 Deep Image Classification Models." Proceedings of the European Conference on Computer Vision (ECCV). 2018.
>
> Elena Voita and Ivan Titov. Information-theoretic probing with minimum description length. In Proceedings of the 2020 Conference on Empirical Methods in Natural Language Processing (EMNLP), pp. 183–196, 2020.
>
> Whitney, William F., et al. "Evaluating representations by the complexity of learning low-loss predictors." arXiv preprint arXiv:2009.07368 (2020).
>
> McAllester, David, and Karl Stratos. "Formal limitations on the measurement of mutual information." International Conference on Artificial Intelligence and Statistics. PMLR, 2020.
>
> Radford, Alec, et al. "Learning transferable visual models from natural language supervision." International Conference on Machine Learning. PMLR, 2021.
>
> Okolo, Gabriel Iluebe, Stamos Katsigiannis, and Naeem Ramzan. "IEViT: An enhanced vision transformer architecture for chest X-ray image classification." Computer Methods and Programs in Biomedicine 226 (2022): 107141.
>
> Tummala, Sudhakar, et al. "Classification of Brain Tumor from Magnetic Resonance Imaging Using Vision Transformers Ensembling." Current Oncology 29.10 (2022): 7498-7511.
>
> Caron, Mathilde, et al. "Emerging properties in self-supervised vision transformers." Proceedings of the IEEE/CVF International Conference on Computer Vision. 2021.
>
> Chen, Ting, et al. "Big self-supervised models are strong semi-supervised learners." Advances in neural information processing systems 33 (2020): 22243-22255.
>
> Wu, Xiongwei, et al. "A large-scale benchmark for food image segmentation." Proceedings of the 29th ACM International Conference on Multimedia. 2021.

---

### Author Response · Authors · 2022-11-18
**Gentle Reminder from Authors**

Dear Reviewers,

Thank you very much for your constructive comments. In the rebuttal phase, we have tried our best to make point-to-point responses based on your comments and improve our submission accordingly. Considering the deadline of Discussion Stage 1 is approaching, we would like to send a reminder to take a look at our responses. Please let us know if there is anything else we can do for our paper, we are happy to answer any follow-up questions!

Thanks,

Paper 4300 Authors

---

### Decision · Program_Chairs · 2023-01-20

**Decision:**

Reject

**Justification For Why Not Higher Score:**

Reviewers remain unconvinced of the significance of the proposed SynBench score's adversarial robustness perspective.

**Justification For Why Not Lower Score:**

N/A

**Metareview: Summary, Strengths And Weaknesses:**

The submission introduces a task-agnostic and model-agnostic framework called SynBench to measure the quality of pretrained representations. It does so by scoring the learned representations' accuracy-robustness tradeoff on a synthetic Gaussian mixture problem (and more specifically how it compares against the analytical tradeoff in the input space). The paper shows that its proposed score correlates well with established metrics and observations.

All reviewers agree on the importance and value of task-agnostic evaluation, especially in the context of large foundation models deployed on multiple downstream tasks. Reviewer xsCU also notes the submission's solid theoretical foundation, and Reviewer REYd notes its exhaustive discussion of related work. Most concerns center around novelty, significance, and comparison to previous work:

- Reviewer REYd is wondering why the model's behavior on images synthesized from a Gaussian mixture would inform its behavior on real-world images.
- Reviewers REYd and Lj97 question the choice of adversarial robustness as a proxy for "representation quality".
- Reviewer REYd finds it difficult to situate the usefulness of the proposed score in comparison to related work (like mutual information or minimum description length). Moreover, there does not appear to be any strong positive correlation between the proposed score and actual tradeoff performances of models in Table 3, and more architectures and datasets would need to be investigated in order to confidently claim that the score is task-agnostic and applies to a wide range of pretrained models (a concern shared by Reviewers xsCU and Lj97).

While the authors' response adds numerous other experiments and makes a serious attempt to address all reviewers' concern, reviewers remain unconvinced of the significance of the proposed SynBench score's adversarial robustness perspective. For instance, Reviewer REYd notes that the experiments focus on OOD robustness, which doesn't quite square with the adversarial robustness perspective. Finally, reviewers have a different reading for the ranking correlation table presented in response to Reviewer REYd and are not convinced that it clearly demonstrates that the SynBench score reliably correlates with robustness on real data.

**Summary Of Ac-Reviewer Meeting:**

N/A